# PI3Kδ hyper-activation promotes development of B cells that exacerbate *Streptococcus pneumoniae* infection in an antibody-independent manner

Anne-Katrien Stark [1,2], Anita Chandra[1,2,3,4], Krishnendu Chakraborty[1,3], Rafeah Alam[1], Valentina Carbonaro[1], Jonathan Clark[5], Srividya Sriskantharajah[6], Glyn Bradley[7], Alex G. Richter[8,9], Edward Banham-Hall[1,3,4], Menna R. Clatworthy[10], Sergey Nejentsev[3], J. Nicole Hamblin[6], Edith M. Hessel[6], Alison M. Condliffe[11] & Klaus Okkenhaug [1,2]

*Streptococcus pneumoniae* is a major cause of pneumonia and a leading cause of death worldwide. Antibody-mediated immune responses can confer protection against repeated exposure to *S. pneumoniae*, yet vaccines offer only partial protection. Patients with Activated PI3Kδ Syndrome (APDS) are highly susceptible to *S. pneumoniae*. We generated a conditional knock-in mouse model of this disease and identify a CD19$^+$B220$^-$ B cell subset that is induced by PI3Kδ signaling, resides in the lungs, and is correlated with increased susceptibility to *S. pneumoniae* during early phases of infection via an antibody-independent mechanism. We show that an inhaled PI3Kδ inhibitor improves survival rates following *S. pneumoniae* infection in wild-type mice and in mice with activated PI3Kδ. These results suggest that a subset of B cells in the lung can promote the severity of *S. pneumoniae* infection, representing a potential therapeutic target.

[1] Laboratory of Lymphocyte Signalling and Development, Babraham Institute, Cambridge CB21 3AT, UK. [2] Division of Immunology, Department of Pathology, University of Cambridge, Cambridge CB2 1QP, UK. [3] Department of Medicine, University of Cambridge, Cambridge CB2 0QQ, UK. [4] Cambridge University Hospitals NHS Trust, Hills Road, Cambridge CB2 0QQ, UK. [5] Biological Chemistry Laboratory, Babraham Institute, Cambridge CB21 3AT, UK. [6] Refractory Respiratory Inflammation Discovery Performance Unit, Respiratory Therapy Area, GlaxoSmithKline, Stevenage SG1 2NY, UK. [7] Computational Biology and Statistics, Target Sciences, GlaxoSmithKline, Stevenage SG1 2NY, UK. [8] Department of Immunology, Queen Elizabeth Hospital, Birmingham B15 2TH, UK. [9] Institute of Immunology and Immunotherapy, University of Birmingham, Birmingham B15 2TT, UK. [10] Molecular Immunity Unit, MRC Laboratory of Molecular Biology, University of Cambridge Department of Medicine, MRC Laboratory of Molecular Biology, Cambridge CB2 0QQ, UK. [11] Department of Infection, Immunity and Cardiovascular Diseases, University of Sheffield, Sheffield S10 2RX, UK. These authors contributed equally: Anne-Katrien Stark, Anita Chandra. Correspondence and requests for materials should be addressed to K.O. (email: ko256@cam.ac.uk)

**S**treptococcus pneumoniae is an invasive extracellular bacterial pathogen and is a leading cause of morbidity and mortality. Although S. pneumoniae can cause disease in immunocompetent adults, it commonly colonizes the upper airways without causing disease. The World Health Organization has estimated that there are 14.5 million episodes of severe pneumococcal disease and that 1.6 million people die of pneumococcal disease every year[1]. Despite the implementation of global vaccination programs, S. pneumoniae infection remains a major disease burden[1–3].

Invasive S. pneumoniae infection is a major cause of lower airway infections (pneumonia), sepsis and meningitis. Healthy people at the extremes of age are more susceptible to pneumococcal disease, as are people with chronic obstructive pulmonary disease (COPD), however those at greatest risk are patients with splenic dysfunction or immune deficiency. This increased susceptibility results at least in part from the lack of protective antibodies against conserved protein antigens or against polysaccharides that form part of the pneumococcal capsule[4]. Indeed, the protective role of antibodies in pneumococcal disease is most obvious in individuals with congenital (primary) immunodeficiencies (PIDs). This was first recognized in a patient with X-linked agammaglobulinemia (XLA), a syndrome subsequently shown to be caused by a block in B cell development due to loss-of-function mutations in BTK[5–7]. These patients remain highly susceptible to S. pneumoniae into adulthood, but can be effectively treated by the administration of immunoglobulins from healthy donors.

We and others have recently described cohorts of immune deficient patients with activating mutations in PIK3CD, the gene encoding the p110δ catalytic subunit of phosphoinositide 3-kinase δ (PI3Kδ)[8–10]. PI3Kδ is a lipid kinase that catalyzes the phosphorylation of the phosphatidylinositol-(4,5)-bisphosphate lipid to produce phosphatidylinositol-(3,4,5)-trisphosphate (PIP$_3$). PI3Kδ is expressed in cells of the immune system and regulates many aspects of immune cell signaling, particularly in lymphocytes[11,12]. Activated phosphoinositide 3-kinase δ syndrome (APDS) is a combined immunodeficiency affecting T and B cells. APDS patients suffer from recurrent sinopulmonary infections, with S. pneumoniae being the most commonly isolated pathogen[13]. Eighty-five percent of APDS patients have been diagnosed with pneumonia[14]. APDS patients are also more likely to develop structural lung damage (bronchiectasis) than patients with other PIDs[13]. The mechanism underpinning the increased susceptibility to pneumococcal infection in APDS is unclear[11].

Although APDS patients often lack IgG2, the protection afforded by immunoglobulin replacement therapy is not as robust as that observed in patients with pure antibody deficiencies, suggesting that antibody-independent PI3Kδ-driven mechanisms may be involved[13]. The monogenic nature of APDS allows us to dissect mechanisms of susceptibility to S. pneumoniae infection on cellular and molecular levels, and to determine whether PI3Kδ inhibitors may help reduce the susceptibility to S. pneumoniae. If so, PI3Kδ inhibitors, that are in development for the treatment of inflammatory and autoimmune diseases, might also have wider applications to reduce the pathological consequences of S. pneumoniae infection[15]. In this study, we have explored mechanisms by which PI3Kδ hyperactivation drives susceptibility to S. pneumoniae infection. We found that the administration of the PI3Kδ-selective inhibitor nemiralisib (GSK-22696557)[16,17] reduced the severity of pneumococcal disease in wild-type mice. To investigate this further, we generated a p110δ[E1020K] mouse model that accurately recapitulates the genetics and immunological phenotype of APDS, and displays increased susceptibility to S. pneumoniae infection. We show that this susceptibility segregates with enhanced PI3Kδ signaling in B cells, which exacerbate S. pneumoniae infection at early time points before the adaptive immune response comes into play. Of note, we have identified a previously unappreciated population of CD19[+]B220[−] IL-10-secreting cells that was present in wild-type mice but expanded 10–20-fold in p110δ[E1020K] mice. We demonstrate that nemiralisib reduces the frequency of IL-10-producing B cells in the lung and improves survival of p110δ[E1020K] mice. Similarly, a higher proportion of transitional B cells from APDS patients produced IL-10 and this was reduced by nemiralisib. This study provides new insights into the pathogenesis of the early stages of invasive S. pneumoniae disease and offers the potential of future therapeutic strategy to alleviate the severity of this disease in susceptible patients.

## Results

**Nemiralisib improves S. pneumoniae infection outcome in mice.** Given that APDS patients are more susceptible to S. pneumoniae, we sought to determine whether nemiralisib, an inhaled PI3Kδ inhibitor which is in development for the treatment and prevention of COPD excacerbations[16,17], would alter susceptibility to airway infections. We treated mice with nemiralisib and then infected them intranasally with S. pneumoniae (TIGR4, serotype 4). Nemiralisib-treated mice showed prolonged

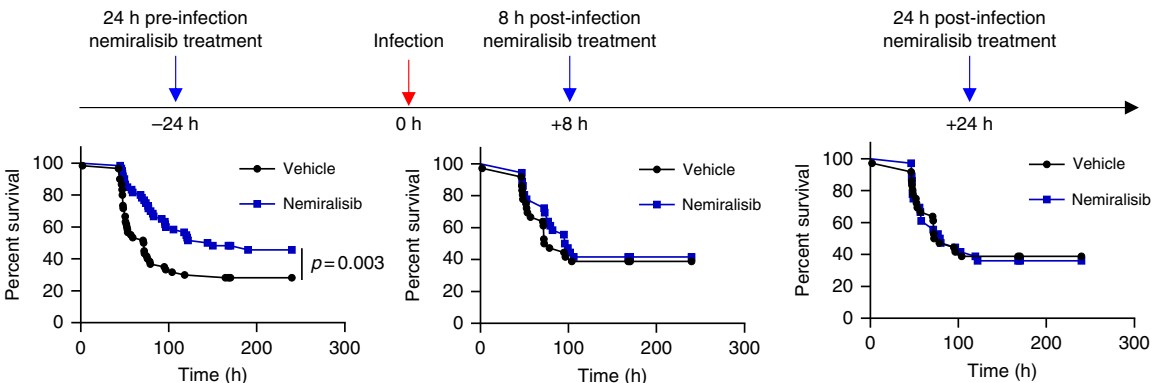

**Fig. 1** Prophylactic, but not therapeutic treatment with the inhaled PI3Kδ inhibitor nemiralisib mitigates disease severity following S. pneumoniae infection in wild-type mice. Wild-type mice were treated twice daily with the inhaled PI3Kδ inhibitor nemiralisib for the duration of the study: when treatment was started 24 h prior to infection with S. pneumoniae serotype 4, TIGR 4, survival rates were improved. When started 8 or 24 h post-infection, the treatment had no effect on survival outcome. (−24 h: data from five independent experiments combined n = 60; +8 h/+24 h: data from three independent experiments combined n = 36; data-points represent individual animals)

survival compared to mice given vehicle control (Fig. 1). This protection was only effective if the drug was administered before and during infection (Fig. 1). By contrast, nemiralisib administration 8 or 24 h post-infection had no impact on survival of the mice. These data suggest that PI3Kδ modulates the immune response during early *S. pneumoniae* infection, either by inhibiting protective immunity, or by promoting an adverse response.

**Hyperactive PI3Kδ signaling alters lymphocyte development.** We generated a conditional knock-in mouse harboring mutation E1020K in the *Pik3cd* gene that is equivalent to the most common APDS-causing mutation E1021K in humans (Supplementary Fig. 1). These mice were subsequently crossed with different Cre-expressing lines to either generate germline mice where p110δ$^{E1020K-GL}$ is expressed in all cells (p110δ$^{E1020K-GL}$) or selectively in B cells using *Mb1*$^{Cre}$ (p110δ$^{E1020K-B}$), in T cells using *Cd4*$^{Cre}$ (p110δ$^{E1020K-T}$) or myeloid cells using *Lyz2*$^{Cre}$ (p110δ$^{E1020K-M}$). We studied p110δ$^{E1020K}$ mice in comparison with wild-type and p110δ$^{D910A}$ mice that have catalytically inactive p110δ[18].

Initially, we tested if p110δ$^{E1020K}$ mice have increased PI3Kδ activity and display the characteristic immunological phenotype of APDS. Biochemical analyzes of B cells and T cells from p110δ$^{E1020K-GL}$ mice confirmed that the kinase is hyperactive (Fig. 2). Measurements of PIP$_3$ in T cells showed that p110δ$^{E1020K}$ is about six times as active as the wild-type kinase following stimulation with anti-CD3 and anti-CD28, but with no evidence for increased basal activity (Fig. 2a). In B cells, p110δ$^{E1020K}$ led to increased basal PIP$_3$ levels, but it was further increased only about two-fold compared to wild-type mice after stimulation with anti-IgM (Fig. 2b). This pattern resembles results found in patients with APDS[9]. In wild-type and p110δ$^{E1020K-GL}$ cells, the PI3Kδ-selective inhibitor nemiralisib reduced PIP$_3$ to the background level observed in p110δ$^{D910A}$ cells, which, as expected, were insensitive to nemiralisib (Fig. 2a, b).

PIP$_3$ binds to the protein kinase AKT, supporting its phosphorylation on Thr308 and subsequent activation. Western blotting of purified p110δ$^{E1020K-GL}$ T cells showed increased AKT phosphorylation following stimulation with anti-CD3 and anti-CD28 antibodies compared to wild-type cells, whereas AKT phosphorylation in p110δ$^{D910A}$ T cells was below the limit of detection (Fig. 2c). In B cells, both basal and anti-IgM-induced phosphorylation of AKT were elevated in p110δ$^{E1020K-GL}$ cells, while strongly diminished in p110δ$^{D910A}$ B cells (Fig. 2d). The phosphorylation of ERK and the AKT effector proteins, FOXO and S6, were similarly affected. All phosphorylation events in wild-type and p110δ$^{E1020K-GL}$ cells were reduced to the levels observed in p110δ$^{D910A}$ cells by inhibition with nemiralisib. As expected, p110δ protein expression was not affected by the E1020K or D910A mutations (Fig. 2c, d).

Germline p110δ$^{E1020K-GL}$ mice had near normal numbers of myeloid cells in the bone marrow and spleen. In the bone marrow we observed a significant B cell lymphopenia that was associated with a block in B cell development between Pro-B and Pre-B cells and did not extend to the spleen. Although these mice had normal numbers of splenic B cells, there was an increased proportion of marginal zone B and B1 cells with an altered distribution of transitional B cells (Fig. 3). The thymus of p110δ$^{E1020K-GL}$ mice was normal except for a mild reduction in single positive CD8$^+$ T cells. In the spleens and lymph nodes there were increased proportions of activated/memory T cells identified by high CD44 expression and low CD62L expression, and increased numbers of Foxp3$^+$ T regulatory cells

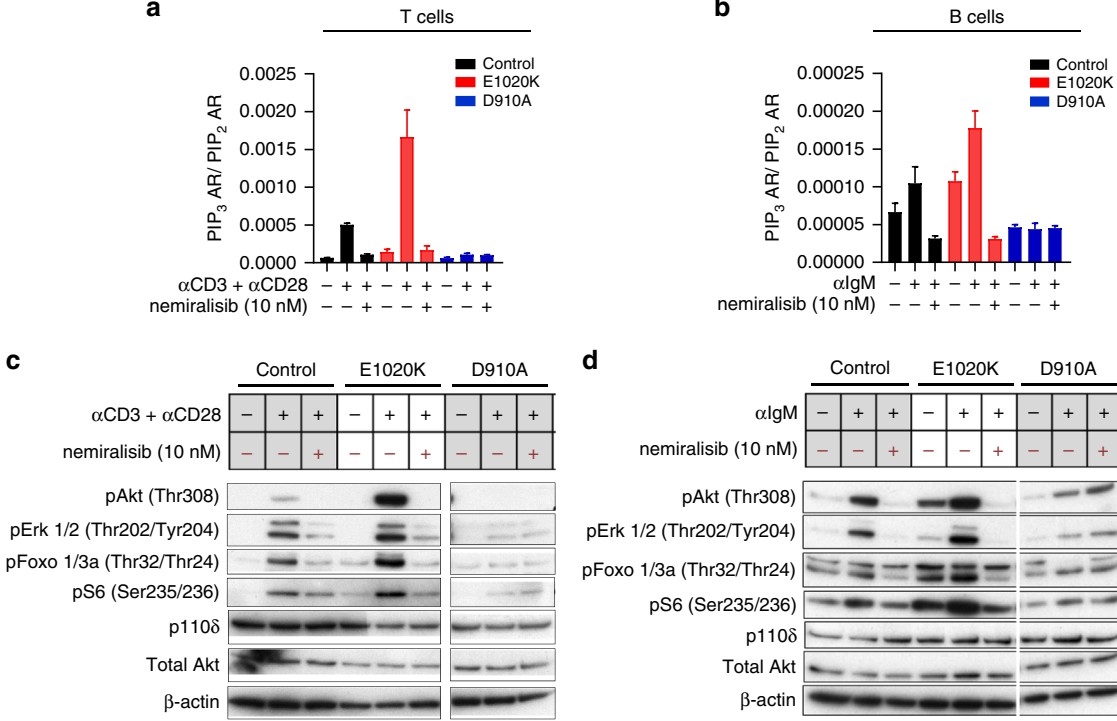

**Fig. 2** PI3Kδ hyper-activation leads to increased PIP$_3$ and pAKT levels that can be reduced using a selective PI3Kδ inhibitor. **a** PIP$_3$ levels in purified T cells from wild-type, p110δ$^{E1020K}$ and p110δ$^{D910A}$ mice, unstimulated or stimulated with anti-CD3 and anti-CD28 in the presence or absence of the selective PI3Kδ inhibitor, nemiralisib (mean ± SD; n = 2–3). **b** PIP$_3$ levels in purified B cells from wild-type, p110δ$^{E1020K}$ and p110δ$^{D910A}$ mice unstimulated, or stimulated with anti-IgM in the presence or absence of nemiralisib (mean ± SD n = 2–6). **c** Western blots of purified T cells from wild-type, p110δ$^{E1020K}$ and p110δ$^{D910A}$ mice, unstimulated or stimulated with anti-CD3 and anti-CD28 in the presence or absence of nemiralisib. **d** Western blots of purified B cells from wild-type, p110δ$^{E1020K}$ and p110δ$^{D910A}$ mice, unstimulated or stimulated with anti-IgM in the presence or absence of nemiralisib. (Representative of two independent experiments; data-points represent individual animals)

(Supplementary Fig. 2). These T cell and B cell phenotypes were recapitulated in p110$\delta^{E1020K-T}$ and p110$\delta^{E1020K-B}$ mice, respectively (Supplementary Figs 3 and 4). The reciprocal effects of the inhibitory D910A and activating E1020K mutations on specific cell subsets demonstrate the pivotal role of PIP$_3$ during lymphocyte development and highlight the importance of a tight control of PI3Kδ activity.

Analysis of serum immunoglobulins showed that p110$\delta^{E1020K-GL}$ mice had elevated levels of IgG1 and IgG2b and a trend towards increased levels of IgG2c, IgA, and IgE isotypes compared to wild-type mice. There was also a trend to hyper IgM in p110$\delta^{E1020K-GL}$ mice as is frequently observed in APDS patients[9,10,13], whereas p110$\delta^{D910A}$ mice were antibody deficient (Supplementary Fig. 5). The level of serum IgG3, which has been shown to be protective

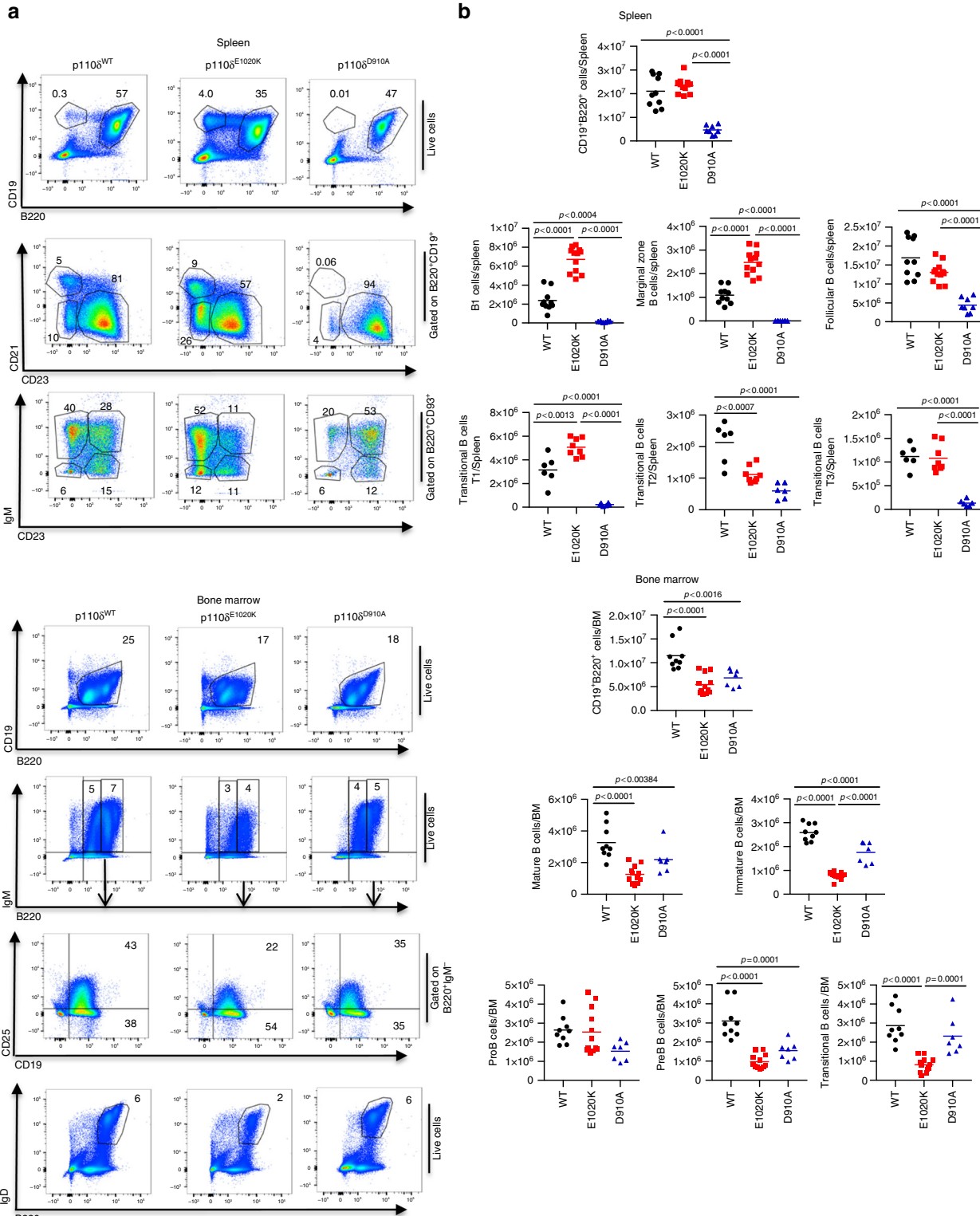

against *S. pneumoniae*[18], was comparable in p110δ[E1020K-GL] and wild-type mice (Supplementary Fig. 5).

**PI3Kδ hyperactivation drive susceptibility to *S. pneumoniae*.** Given that the immunological phenotype of p110δ[E1020K-GL] mice strongly resembled that of APDS patients, we sought to determine whether these mice recapitulate the increased susceptibility to *S. pneumoniae* observed in APDS patients. We infected p110δ[E1020K-GL], p110δ[D910A], and p110δ[WT] mice with *S. pneumoniae* (TIGR4 serotype 4) and followed their survival for 10 days (Fig. 4a). Interestingly, p110δ[D910A] mice did not show increased susceptibility to *S. pneumoniae*. By contrast, p110δ[E1020K-GL] mice showed accelerated disease onset and increased mortality (Fig. 4a). Natural antibodies against phosphorylcholine (PC) can offer protection against infection with encapsulated bacteria, including *S. pneumoniae*[19,20]. We found that p110δ[D910A] mice lacked anti-PC antibodies, presumably because of the absence of B1 and MZ B cells which are the major source of natural antibodies[18,21–23]. By contrast, anti-PC antibodies in the serum from p110δ[E1020K] mice were similar (IgM) or elevated (IgG) (Fig. 4b, c) compared to wild-type mice. Therefore, susceptibility to *S. pneumoniae* in p110δ[E1020K] mice cannot be explained by a failure to produce natural antibodies against conserved bacterial epitopes.

In order to determine the cell type responsible for increased susceptibility to *S. pneumoniae*, we next infected the lineage-restricted p110δ[E1020K-B], p110δ[E1020K-T] and p110δ[E1020K-M] mice. Myeloid expression of p110δ[E1020K] had no effect on the course of *S. pneumoniae* infection (Fig. 4d), whereas expression of p110δ[E1020K] in T cells was protective (Fig. 4e). Only the p110δ[E1020K-B] mice replicated the increased susceptibility of the p110δ[E1020K-GL] mice to *S. pneumoniae* (Fig. 4f). Furthermore, transfer of p110δ[E1020K-B] bone marrow into irradiated RAG2[−/−] recipients also conferred increased susceptibility to infection, which was only partially rescued by co-transferring wild-type and p110δ[E1020K-B] bone marrow at a 1:1 ratio (Fig. 4g). These results indicate that B cells drive the increased susceptibility to *S. pneumoniae* infection in p110δ[E1020K-GL] mice in an immune-dominant manner.

**B cell mediated susceptibility is antibody-independent.** In order to further investigate antibody-mediated protection in the context of PI3Kδ hyper-activation, we immunized mice with Pneumovax, a 23-valent polysaccharide vaccine[24]. Following infection with *S. pneumoniae*, wild-type mice were completely protected by this vaccination protocol, while p110δ[E1020K-GL] mice were only partially protected, to a level similar to that in non-immunized wild-type mice. By contrast, p110δ[D910A] mice did not benefit from vaccination (Fig. 5a, b). Interestingly, p110δ[E1020K-GL] mice and wild-type mice produced a similar antibody response, in contrast to p110δ[D910A] mice that showed no response (Fig. 5c). These data indicate that Pneumovax vaccination clearly protects against *S. pneumoniae*, as shown in immunized wild-type mice. Despite this protection p110δ[E1020K] mice remained significantly more susceptible to *S. pneumoniae* infection. Together, these data suggest

that B cells can affect susceptibility to *S. pneumoniae* by a mechanism that is at least in part antibody-independent.

To determine more definitively whether B cells can be pathogenic in the context of *S. pneumoniae* infection, we infected wild-type and *Ighm*[tm1] (μMT) mice which lack mature B cells[25]. Strikingly, *Ighm*[tm1] mice showed reduced susceptibility to *S. pneumoniae* infection, delaying disease onset from ~2 days in wild-type mice to ~5 days in *Ighm*[tm1] mice (Fig. 6). Although survival in *Ighm*[tm1] mice was increased up to 30 days post infection compared to wild-type mice, CFU counts from the lungs of mice surviving to this time-point indicates that, despite appearing clinically healthy, 41% (7/17) of *Ighm*[tm1] mice failed to clear the infection compared to 100% clearance in wild-type mice (Fig. 6). Taken together, these results suggest that during early time-points in the local infected environment B cells can be pathogenic, while at later stages they are required to prevent chronic infection.

Increased susceptibility to *S. pneumoniae* could either be due to uncontrolled bacterial proliferation or be caused by an aberrant immune response to the pathogen. The bacterial titers from lungs and spleens of wild-type, p110δ[D910A] and p110δ[E1020K-GL] mice were similar at 24 h post-infection, suggesting that the different susceptibilities did not correlate with different abilities to control bacterial outgrowth during the early phase of infection (Fig. 7a). We observed a trend towards increased levels of TNFα, IL-6, IL-1β, and IL-1α in the lung tissue of p110δ[E1020K-GL] mice at 24 h post-infection (Fig. 7b). Consistent with this, the levels of TNFα, IL-6, and IL-1β were reduced in the lungs of wild-type mice treated with nemiralisib prior to *S. pneumoniae* infection (Fig. 7c), indicating that, while PI3Kδ signaling affect the amount of pro-inflammatory cytokines produced in response to infection, the increase in response to PI3Kδ hyper-activation is modest.

**CD19[+]B220[−] B cells produce high levels of IL-10.** We hypothesized that increased susceptibility to *S. pneumoniae* infection can be mediated by a specific subpopulation of B cells. Therefore, we studied the B cell compartment in various tissues and found an atypical population of CD19[+]B220[−] B cells that was rare in wild-type mice, but significantly increased in p110δ[E1020K-GL] mice and was absent in p110δ[D910A] mice (Fig. 8a). In the spleen and bone marrow, there was a 10-fold and 5-fold increase, respectively, in the numbers of CD19[+]B220[−] B cells in p110δ[E1020K-GL] mice compared to wild-type mice (Fig. 8a). Analysis of B cell conditional p110δ[E1020K-B] mice show a similar expansion of B220[−] B cells in the spleen (Fig. 8a), indicating that the development of this population in response to PI3Kδ hyperactivation is B-cell intrinsic. Infection with *S. pneumoniae* did not induce further expansion of CD19[+]B220[−] B cells in the lungs or other tissues examined 24 h post-infection (Fig. 8b).

To ascertain if these cells were also present in the lungs and to distinguish resident from circulating cells, we labeled circulating leukocytes in wild-type and p110δ[E1020K-GL] mice by intravenous injection of biotin-conjugated anti-CD45. We then stained the lung homogenate with fluorochrome-conjugated anti-mouse

**Fig. 3** Phenotype of B cells in p110δ[E1020K-GL] mice. **a** B cell subsets in the spleen and bone marrow from wild-type, p110δ[E1020K-GL] and p110δ[D910A] mice were analyzed by flow cytometry and representative pseudocolor plots with the mean cell proportion are shown. **b** In the spleen, the number of B1, marginal zone (MZ) and T1 transitional cells B cells were increased in p110δ[E1020K-GL] mice while follicular B cell numbers were normal. These populations were reduced in p110δ[D910A] mice. Analysis of the bone marrow showed normal pro-B cell numbers in p110δ[E1020K-GL] mice with a reduced number of pre-B cells, immature, transitional and mature B cells. Populations of cells are described as follows: Splenic B cells: Total B cells CD19[+]B220[+], B1 cells CD19[+]B220[+]CD23[−]CD21[−], Follicular B cells CD19[+]B220[+]CD23[+]CD21[+], Marginal zone B cells CD19[+]B220[+]CD23[−]CD21[−], Transitional T1 B cells B220[+]CD93[+]IgM[+]CD23[−], Transitional T2 B cells B220[+]CD93[+]IgM[+]CD23[+], Transitional T3 B cells B220[+]CD93[+]IgM[−]CD23[−]; Bone marrow B cells - Immature B cells CD19[+]B220[lo]IgM[+], Mature B cells CD19[+]B220[hi]IgM[+], Pro-B cells B220[+]IgM[−]CD19[+]CD25[−], Pre-B cells B220[+]IgMCD19[+]CD25[+], Transitional B cells B220[+]IgD[+]. (Mean cell numbers are shown; combined data from two independent experiments, wild-type *n* = 10; p110δ[E1020K-GL]*n* = 12; p110δ[D910A]*n* = 7. Data-points represent individual animals)

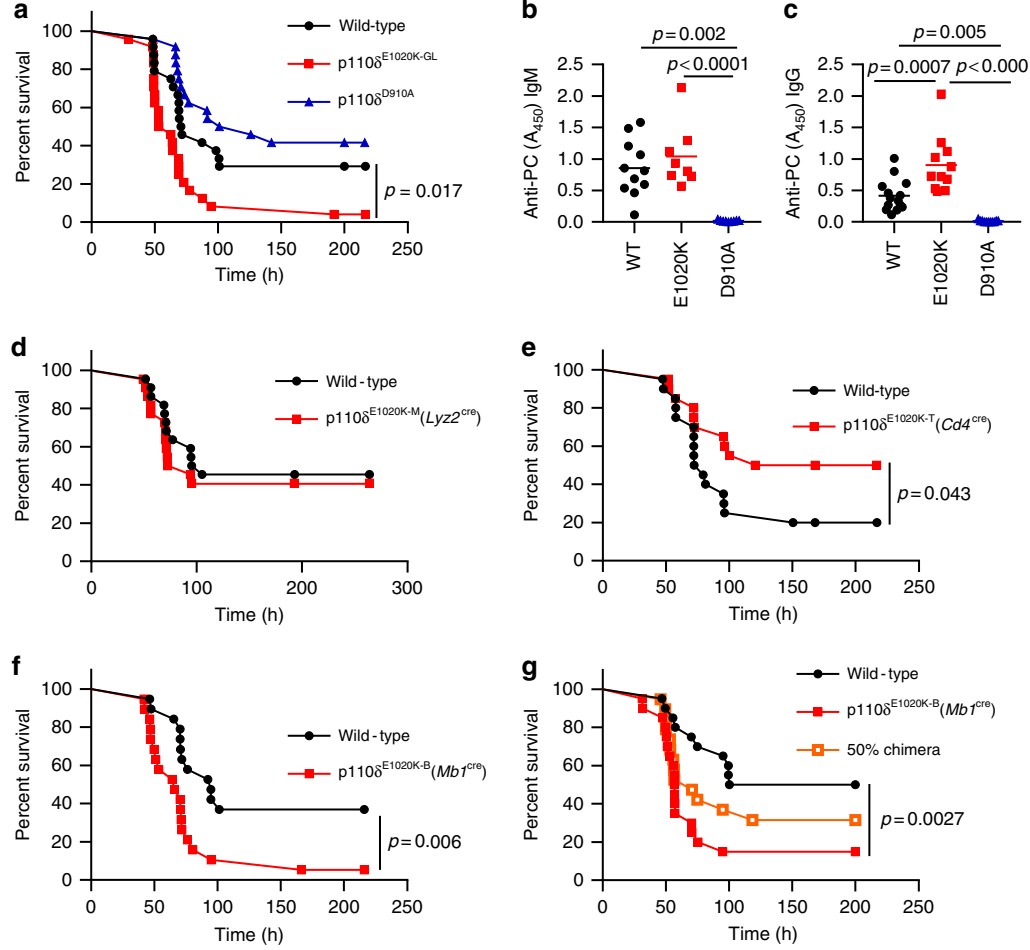

**Fig. 4** PI3Kδ hyper-activation leads to increased susceptibility to *S. pneumoniae* infection. **a** Germline p110δ[E1020K-GL] mice show accelerated disease development and significantly increased mortality compared to control mice in response *S. pneumoniae* infection, while kinase dead p110δ[D910A] mice do not respond differently to wild-type mice. **b**–**c** Naïve PI3Kδ[E1020K] mice produce normal levels of anti-phosphorylcholine IgM and significantly higher levels of anti-PC IgG, while PI3Kδ[D910A] mice produce no natural antibody. **d**–**f** The p110δ[E1020K] mutation was introduced conditionally into myeloid cells, T cells and B cells by crossing onto *Lyz2*[cre], *Cd4*[cre] and *Mb1*[cre] lines respectively. The myeloid conditional mutation did not affect survival following *S. pneumoniae* infection in p110δ[E1020K-M] mice (**d**), while T cell conditional p110δ hyper-activation led to improved survival in p110δ[E1020K-T] mice (**e**). Introducing the p110δ[E1020K] mutation specifically in B cells (p110δ[E1020K-B] mice) replicated the increased susceptibility to *S. pneumoniae* seen in p110δ[E1020K-GL] mice (**f**). **g** Transfer of p110δ[E1020K-B] bone marrow into irradiated RAG2[−/−] recipients also conferred increased susceptibility to *S. pneumoniae* infection compared to recipients receiving wild-type bone marrow, and this phenotype was not fully rescued in a 50% BM chimera. (Data-points represent individual animals; data from 2 independent experiments combined. **a** n = 24; **b**, **c** mean antibody levels shown, WT n = 11, E1020K n = 8, D901A n = 9; **d** n = 22; **e** n = 20; **f** n = 20; **g** n = 20)

CD45 and streptavidin, and used flow cytometry to distinguish tissue resident leukocytes from those present in the lung capillaries. We found a significant increase in the proportion and number of tissue-resident, but not circulating, B cells in p110δ[E1020K-GL] mice compared to wild-type mice. There was an increase in both CD19[+]B220[+] and CD19[+]B220[−] cells among the tissue-resident B cells which demonstrates that both populations can take residence in the lung (Supplementary Fig. 6).

Given that B cells from p110δ[E1020K-GL] mice drive susceptibility to *S. pneumoniae* in an antibody-independent manner, we sought to look at other properties of B cells. We found lower IL-10 protein levels and a significant 10-fold decrease in IL-10 mRNA expression at 24 h post-infection in whole lung homogenates from wild-type mice treated with nemiralisib (Fig. 7c, d), indicating that PI3Kδ signaling can regulate IL-10 levels in the lung post infection. IL-10 is an important immune-regulatory cytokine known to affect the course of *S. pneumoniae* infection[26,27] and therefore increased IL-10 production could explain the B cell-dependent but antibody-independent effects

seen in p110δ[E1020K-GL] mice. To investigate this further, we crossed the p110δ[E1020K-GL] and p110δ[D910A] mice with a highly sensitive IL-10 reporter (*Il10*[ITIB]) mouse[28].

At 24 h post *S. pneumoniae* infection, the frequencies of the IL-10-producing B cells were significantly increased in the lungs of p110δ[E1020K-GL]*Il10*[ITIB] mice compared to wild-type *Il10*[ITIB] mice (Fig. 7e). In addition, IL-10-producing B cells were barely detected in lungs from p110δ[D910A] mice (Fig. 7e). Frequencies of the IL-10-producing CD11b[+] myeloid, T and NK cells in lungs were variable, but not consistently increased in p110δ[E1020K-GL] mice (Fig. 7e). Further analysis of the CD19[+] B cell subset isolated from lungs and spleens at 24 h post *S. pneumoniae* infection showed that the proportion of IL-10-producing cells was increased among the B220[−] B cell subset as compared to the B220[+] B cell subset in p110δ[E1020K-GL] mice (5% B220[+] vs. 68% B220[−]) and wild-type mice (2% B220[+] vs. 55% B220[−]) (Fig. 8c). Furthermore, the proportion and absolute number of IL-10-producing CD19[+]B220[−] cells and CD19[+]B220[+] cells were increased in p110δ[E1020K-GL] mice compared to wild-type mice,

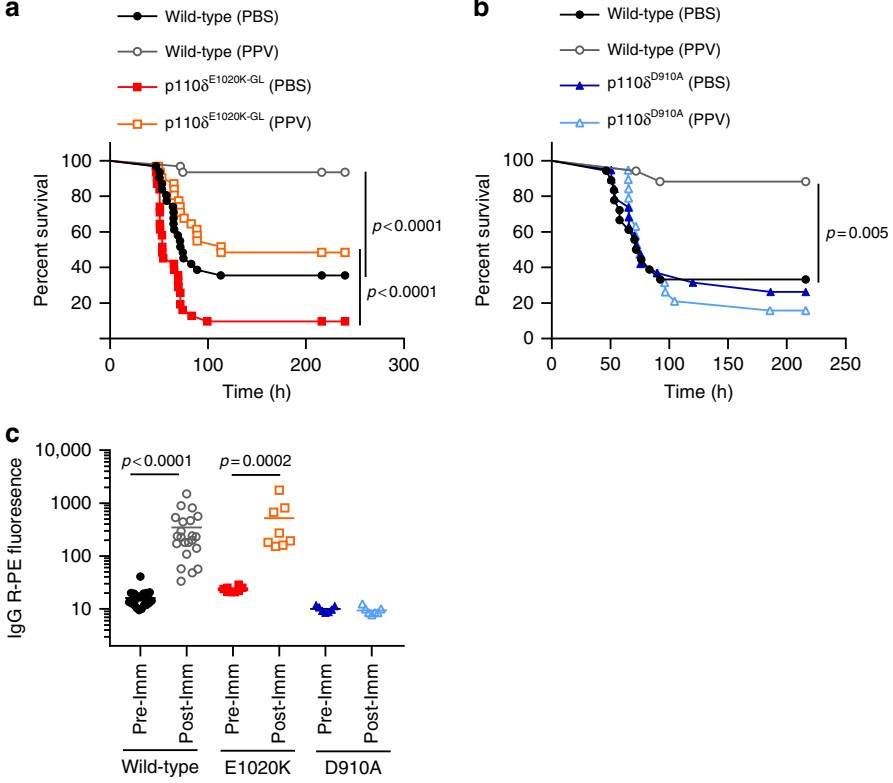

**Fig. 5** PI3Kδ$^{E1020K-GL}$ mice respond to Pneumovax and have normal antibody levels. **a**, **b** Pneumovax (T-independent pneumococcal polysaccharide vaccine, PPV) partially protects p110δ$^{E1020K-GL}$ mice against infection (**a**), while p110δ$^{D910A}$ are not protected (**b**). **c** p110δ$^{E1020K-GL}$ mice produce normal levels of total IgG in response to Pneumovax immunization (anti-pneumococcal serotype 4 shown) while p110δ$^{D910A}$ mice do not respond to vaccine. (Data-points represent individual animals; **a** results from three independent studies combined, $n = 30$; **b** results from two independent studies combined, $n = 19$; **c** results from two independent studies combined: mean antibody levels shown; WT $n = 22$; E1020K $n = 8$; D910A $n = 6$)

while such IL-10-producing cells were virtually absent in p110δ$^{D910A}$ mice (Fig. 8c). This indicates that CD19$^+$B220$^-$ cells are the predominant population of B cells that produce IL-10 in a PI3Kδ-dependent manner.

Previously, B cells that produce IL-10 have been termed B regulatory cells (Bregs)[29]. We sought to compare the phenotype of CD19$^+$B220$^-$ IL-10-producing B cells with conventional IL-10-producing CD19$^+$B220$^+$ Bregs described previously (Fig. 9). Following in vitro stimulation for 5 h and analysis of cell surface markers and intracellular IL-10 we found that there were clear phenotypic differences between conventional B220$^+$ Breg cells and the B220$^-$ B cells we describe here, including the differential expression of CD43 and IgM (Fig. 9). The lack of B220 expression and low IgM expression differentiate the CD19$^+$B220$^-$ IL-10-producing B cells from conventional B1 cells[30].

**Nemiralisib reduces mortality in infected p110δ$^{E1020K}$ mice.** Given that increased PI3Kδ activity in mice leads to increased mortality after *S. pneumoniae* infection we explored if nemiralisib would provide protection. Treatment of p110δ$^{E1020K-GL}$ mice with inhaled nemiralisib 24 h prior to infection led to a 20% increase in survival. While nemiralisib treatment did not affect the numbers of CD19$^+$B220$^-$ B cells, the proportion of IL-10 producing CD19$^+$B220$^-$ B cells in the lungs were reduced in comparison to non-treated mice (Fig. 10a). This was in keeping with our prior observations that nemiralisib treatment reduced IL-10 mRNA levels in the lung tissue of *S. pneumoniae* infected wild-type mice (Fig. 7d), suggesting a role for IL-10-producing CD19$^+$B220$^-$ B cells in the PI3Kδ-dependent susceptibility to infection. Homozygous *Il10*$^{ITIB}$ mice are hypomorphic for IL-10

production. We exploited this property to assess if IL-10 depletion would result in improved survival in p110δ$^{E1020K}$ or WT mice. Interestingly, systemic reduction of IL-10 in homozygous *Il10*$^{ITIB}$ mice did not improve disease outcome (Supplementary fig. 7B). Furthermore, local depletion of IL-10 in the lung through intranasal administration of anti-IL-10 did not improve disease outcome in p110δ$^{E1020K}$ mice (Supplementary fig. 7D). These data suggest that depletion of IL-10 is not sufficient to revert they p110δ$^{E1020K}$ susceptibility phenotype. It is possible that there may be a very specific role for IL-10 produced by the B220$^-$ B cells related to the location of these cells in the lung that we could not attenuate specifically using available tools and reagents. There may also be other cytokines or secreted proteins that are co-regulated with IL-10 in CD19$^+$B220$^-$ B cells that are also required. Indeed, Fig. 7d shows a number of cytokines that were reduced by nemiralisib treatment of infected mice which have the potential to affect the local response to infection. Further characterization of this B cell subset may reveal such additional mechanisms that can be tested experimentally.

**Nemiralisib reduces IL-10 production in APDS patient B cells.** Cohort studies have shown that 75% of patients with APDS have elevated circulating transitional B cells (CD19$^+$IgM$^{++}$CD38$^{++}$)[13]. Other studies have shown that such transitional B cells can produce high levels of IL-10[31,32]. We obtained blood samples from patients with APDS and healthy controls and confirmed that all of these APDS patients had elevated proportions of transitional B cells (Fig. 10b). We isolated PBMCs from two of these APDS patients and three healthy controls and stimulated them for 72 h with anti-CD3 and IL-2[31]. After stimulation we found more IL-10-producing

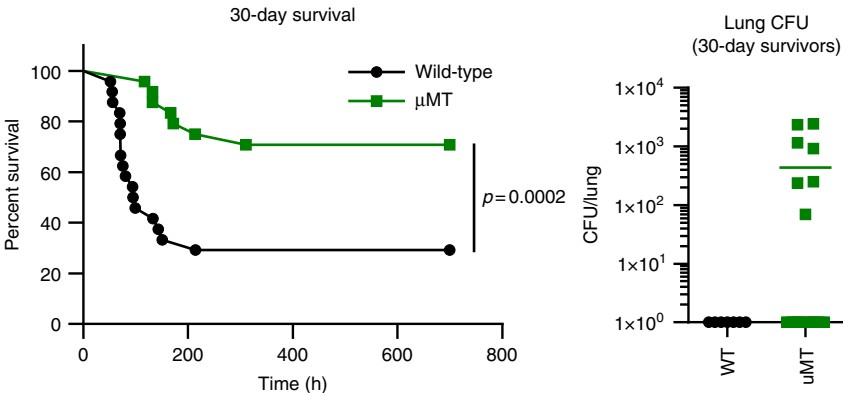

**Fig. 6** B cells could drive increased pathology in response to *S. pneumoniae* infection, independent of bacterial clearance. *Ighm*[tm1] (μMT) mice show delayed disease progression and improved overall survival up to 30 days post *S. pneumoniae* infection, however 41% (7/17) surviving *Ighm*[tm1] mice failed to clear bacteria from the lung tissue at 30 days post-infection compared to 100% clearance in surviving wild-type mice. (Data from 1 study, *n* = 24, mean CFU count shown, data-points represent individual animals)

B cells and more IL-10-producing transitional B cells in PBMCs of APDS patients than in PBMCs of healthy controls, while treatment with nemiralisib effectively suppressed IL-10 production in these cells (Fig. 10c). Therefore, in keeping with reduction of CD19[+]B220[-]IL-10 producing B cells in the mice, nemiralisib can also reduce IL-10 producing human transitional B cells.

## Discussion

Our results show that enhanced PI3Kδ signaling leads to the increased susceptibility to *S. pneumoniae* infection through a B cell-dependent, but antibody-independent mechanism. We found an expansion of an aberrant B cell population which could be equivalent to the elevated transitional B cell population found in APDS patients. These cell populations may have a detrimental effect during the early phase of *S. pneumoniae* infection.

In this study, we show that B cells can increase the susceptibility to *S. pneumoniae* infection during the first few hours after exposure. Mice were more susceptible to *S. pneumoniae* infection when the hyperactive p110δ[E1020K] mutation was expressed in B cells. By contrast, mice expressing the p110δ[E1020K] mutation only in T cells were protected. The basis for the protection remains unknown and is the focus of ongoing study. The p110δ[E1020K] mice have increased numbers of $T_{FH}$ and Treg cells and the conventional T cells have a more activated phenotype. A recent study has documented a population of lung-resident innate Th17 cells that confers protection against re-challenge with *S. pneumoniae*[33]. Whether such a subset is affected by PI3Kδ activity and can also protect against immediate challenge is not known. However, we did not detect increased proportions of Th17 cells after infection. Regardless of the mechanism, the protection offered by T cells was overcome by the adverse effects of B cells in p110δ[E1020K-GL] mice.

We describe a subset of B cells which lack the common B cell marker B220, and whose abundance is correlated with the susceptibility to *S. pneumoniae* infection. Moreover, we show that the development of this CD19[+]B220[-] B cell subset is highly dependent on the level of activity of PI3Kδ. Susceptibility to *S. pneumoniae* infection was not altered in p110δ[D910A] mice as compared to wild-type mice. Kinase-dead p110δ[D910A] mice lack natural and anti-capsular antibodies which may make them more susceptible, however, their lack of CD19[+]B220[-] B cells may counterbalance this antibody deficiency, such that overall the mutation has little net effect on *S. pneumoniae* susceptibility.

A large proportion of the CD19[+]B220[-] B cells produced IL-10. Although increased IL-10 could potentially reduce the production of inflammatory cytokines, we found similar or increased levels of

TNFα, IL-6 and IL-1 in the lungs of p110δ[E1020K] mice and no difference in CFU counts at 24 h post-infection.

One of the defining characteristics of APDS patients is an increased proportion of CD24[+]CD38[+] B cells defined as transitional B cells[9–11,13,14]. Intriguingly, production of IL-10 is a characteristic of these B cells as well. High IL-10 levels in response to secondary *S. pneumoniae* infection following influenza A is associated with increased lethality when compared to a primary *S. pneumoniae* infection, and importantly, the outcome of a secondary *S. pneumoniae* infection is improved by IL-10 neutralization[26,27]. This indicates that, while IL-10 is required for normal immune regulation and resolution of inflammation[34], excess IL-10 during the early stages of *S. pneumoniae* infection could have an acute detrimental effect. This fits with our observation that nemiralisib treatment improved the outcome of mice infected with *S. pneumoniae* and reduced IL-10 mRNA levels in the lung. However, we did not observe improved disease outcome in response to systemic or local IL-10 depletion, indicating that increased IL-10 production is not the sole mechanism driving increased pathology during *S. pneumoniae* infection. Further studies are required to investigate the time and location specific effects of B cell dependent IL-10 production, keeping in mind that B cells have the potential to produce a number of different cytokines (such as IL-6, TNF-α, and IL-35) which were affected by leniolisib treatment and which could affect *S. pneumoniae* susceptibility[35].

We postulate that the transitional B cells which are expanded in APDS patients may not just be precursors to more mature B cells, but also include cells that are functionally equivalent to the CD19[+]B220[-] cells that we have identified in p110δ[E1020K] mice and thus may also contribute to the increased susceptibility to *S. pneumoniae* infection and hence to the high incidence of bronchiectasis characteristic of this disease[14]. In this context, it is interesting that CD19[+]B220[-] B cells had previously been described as immature progenitors in the bone marrow[36], rather than a functional PI3Kδ dependent subset found in peripheral tissues as we describe here. Although the CD19[+]B220[-] B cells we describe resemble B1 cells by some criteria, such as the expression of CD5 and CD43, their lower IgM expression, lack of B220 and distribution pattern suggest that they are related but distinct to conventional B1 and B10 cells[29,30]. A similar subset has previously been described as being dependent on CD19 and increased in the absence of PTEN expression and PTEN haploinsufficiency in humans can lead to an APDS-like syndrome[37–39].

Our results do not question the important role that antibodies play in the protection against *S. pneumoniae* and other encapsulated bacteria. Indeed, in wild-type mice, immunization with

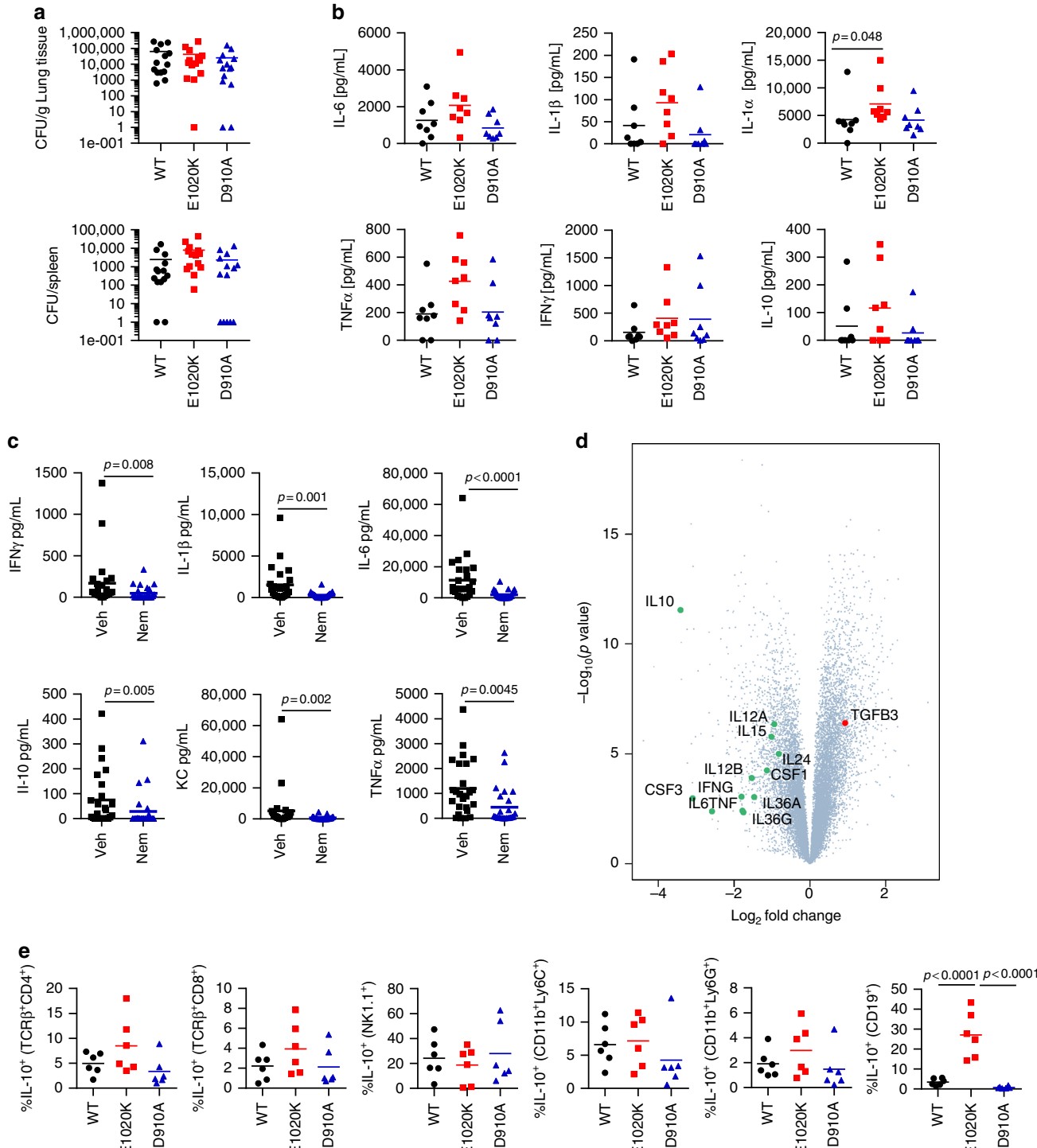

**Fig. 7** PI3Kδ signaling regulates B cell specific IL-10 production in the lung following *S. pneumoniae* infection. **a** 24 h post *S. pneumoniae* infection, lung and spleen CFU counts were similar in wild-type, p110δ$^{E1020K-GL}$ and p110δ$^{D910A}$ mice. **b** 24 h post-infection, cytokine levels in the lung homogenate showed a trend towards increased TNFα, IL-6, and IL-1 in p110δ$^{E1020K}$ mice. **c** 24 h prophylactic treatment with nemiralisib (nem) led to reduced levels of TNFα, IL-6, IL-1β, IFNγ, and IL-10 in the lungs of wild-type mice compared to vehicle control (veh) treated animals at 24 h post-infection. **d** Volcano plot (statistical significance against fold change) of the gene expression changes in response to nemiralisib treatment showed reduced levels of pro-inflammatory cytokines as well as IL-10 at 24 h post infection compared to vehicle control treatment. All genes analyzed are shown (gray dots) with the cytokines of interest labeled and colored; green for those with a negative fold change, red for positive fold change. **e** Analysis of immune cell subsets in *Il10*$^{ITIB}$ reporter mice at 24 h post-infection showed that the proportion of IL-10 producing B cells is significantly increased in p110δ$^{E1020K-GL}$ mice and reduced in p110δ$^{D190A}$ mice compared to wild-type mice, with similar trends in T cells and myeloid cells not reaching significance. (Mean values shown; data-points represent individual animals. **a** Results from two independent studies combined, $n = 14$; **b** representative data from three independent studies $n = 8$; **c** Combined data from four independent experiments $n = 25$; **d** data from 1 study, $n = 6$; **e** representative data from two independent studies $n = 6$)

Pneumovax offered complete protection. Rather, our results highlight a hitherto underappreciated subset of B cells that increase the pathological response shortly after infection. It remains to be determined whether this is a feature of the B cells being lung-resident (and hence among the first cells of the immune system to encounter the invading pathogens) and by which mechanisms B cells either condition lung epithelia to

become more susceptible to invasion of *S. pneumoniae*, reduce the clearance of *S. pneumoniae* by phagocytes and/or increase the collateral adverse effects of the early innate immune response to this pathogen. This is in keeping with the increasing realization that B cells are important sources of different cytokines and can modulate immune responses independently of their capacity to present antigen and produce antibodies[35].

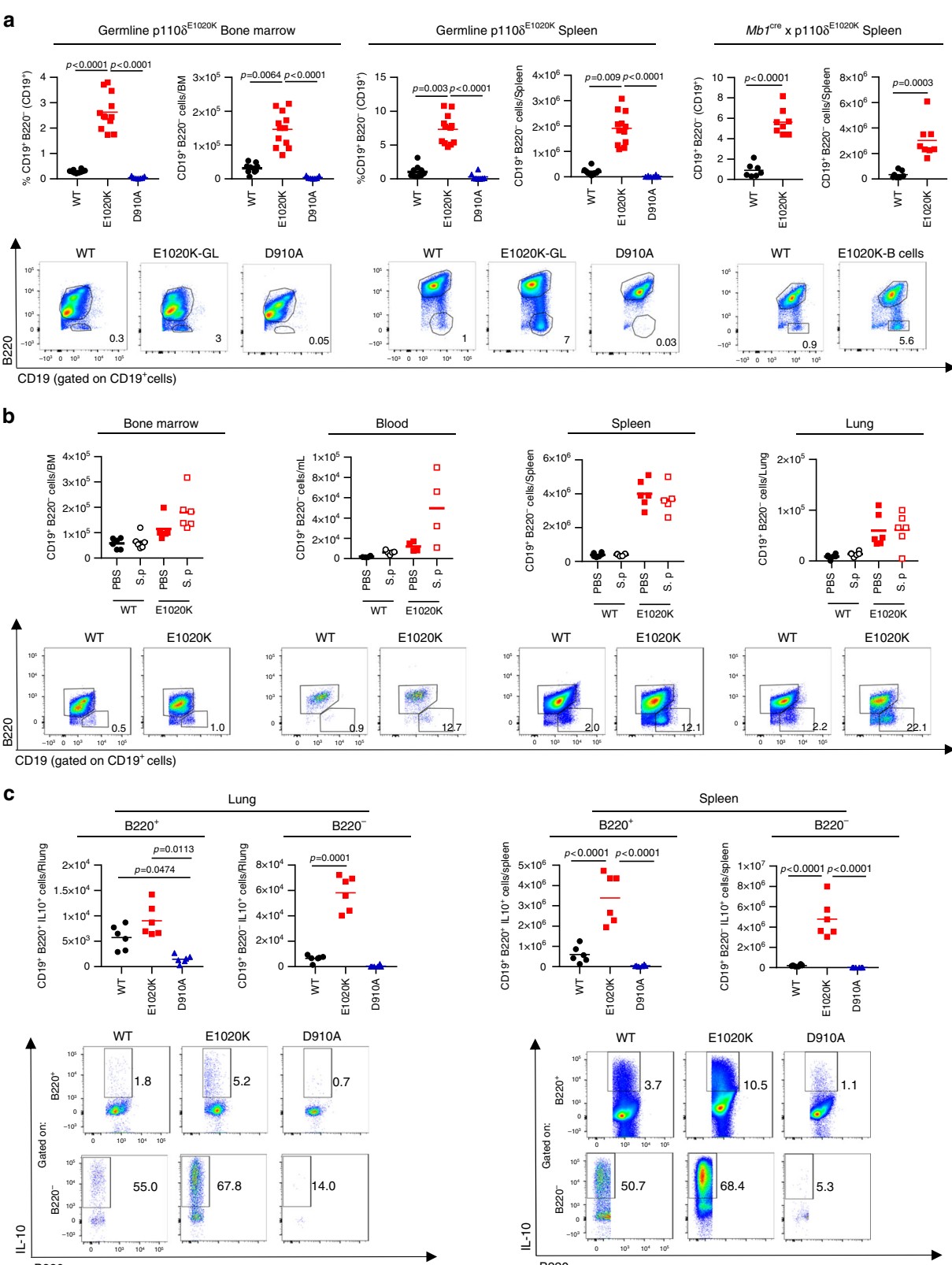

Intriguingly, the inhaled inhibitor nemiralisib offered protection in not just in p110δ[E1020K] but also in wild-type mice infected with *S. pneumoniae*, when administered prior to infection. This result suggests that preventive inhibition of PI3Kδ can be beneficial not only just in the context of APDS, but also in other conditions associated with PI3Kδ hyperactivation, such as COPD[40].

The results presented herein are relevant to our understanding and the potential treatment of patients with APDS. Clinical trials are currently ongoing using both inhaled and systemic PI3Kδ inhibitors (NCT02593539, NCT02435173, and NCT02859727). Systemic use of a PI3Kδ inhibitor has shown promise in reduction of lymphoproliferation and immune cell aberrations in patients with APDS, however, no comments were made regarding recurrent infections or respiratory manifestations[41]. Systemic use of PI3Kδ inhibitors in malignancies have been associated with serious side effects, including colitis[42]. It would therefore be of importance to determine whether inhaled PI3Kδ inhibitors could alleviate the respiratory manifestations of APDS while reducing the risk of the adverse effects associated with systemic inhibition. Moreover, the results may also be relevant to other conditions, such as COPD[16], where PI3Kδ may be activated by non-genetic mechanisms. Overall, our findings suggest that in subjects with increased PI3Kδ activity prophylactic administration of PI3Kδ inhibitors may help alleviate the course of *S. pneumoniae* infection.

## Methods

**Ethics**. Animal experiments were performed according to the Animals (Scientific Procedures) Act 1986, licence PPL 70/7661 and approved by the Babraham Institute Animal Welfare and Ethics Review Body.

Informed consent was obtained from patients and healthy controls. This study conformed to the Declaration of Helsinki and according local ethical review document 12/WA/0148.

**Mouse strains and gene targeting**. The p110δ[E1020K] mice were generated by Ozgene, Australia using homologous recombination in ES cells. A duplicate sequence corresponding to the last coding exon in *Pik3cd* and carrying the E1020K mutation was flanked by loxP sites and inserted 3' to the wild-type sequence. Upon Cre-mediated recombination, the wild-type sequence is replaced by the mutant E1020K sequence. p110δ[D910A], *Cd4*[cre], *Mb1*[cre], *Lyz2*[cre], *Ighm*[tm1] (µMT) and *Il10*[ITIB] mice have been described previously[18,25,28,43–45]. Throughout the study, the p110δ[E1020K] allele was heterozygous whereas the p110δ[D910A] alleles were homozygous. Genotyping was performed by Transnetyx (Cordova, TN). Throughout this study we used male and female animals aged between 8–15 weeks in each experiment, wild-type controls were age and sex matched to genetically modified mice.

**S. pneumoniae stock preparation**. *Streptococcus pneumoniae* (TIGR4, serotype 4 (provided by Professor Jeremy Brown, University College, London)) was grown to mid-log phase ($OD_{500} = 0.5$–$0.7$) in Todd-Hewitt broth (Oxoid) supplemented with 0.5% yeast extract (Oxoid) at 37 °C, 5% $CO_2$. The bacteria were collected by centrifugation, and resuspended in PBS/20% glycerol (Sigma-Aldrich) prior to freezing in liquid $N_2$ and storage at −80 °C. Stocks were assessed for viable CFU counts and homogeneity by plating out serial dilutions of three frozen samples on blood agar plates (LB agar, supplemented with 5% defibrinated sheep blood (Oxoid)) after incubation for 24 h at 37 °C, 5% $CO_2$. *S. pneumoniae* colonies were confirmed by the presence of an α-hemolytic zone and sensitivity to optochin (Sigma-Aldrich). Virulent stocks were maintained by performing an in vivo passage every 6–12 months.

**S. pneumoniae infections**. Frozen stocks were thawed and washed twice by centrifugation in sterile PBS before resuspending at $4 \times 10^7$ CFU/mL in PBS. The suspension was kept on ice at all times, and used for infection within 2 h of thawing (no loss of viability was observed under these conditions). Mice were lightly anaesthetized by inhalation of 3% isoflurane and maintained with 2% isoflurane. Mice (males and females aged 8–12 weeks) were infected intranasally with 50 µL *S. pneumoniae* suspension containing $2 \times 10^6$ CFU. Animals were observed to confirm inhalation of the dose and full recovery from anesthesia. The infection dose was routinely confirmed by plating out serial dilutions of the inoculum on blood agar plates, as described above.

**Survival studies**. Mice were infected with *S. pneumoniae* as described above. Pre-infection body weights were recorded and animals were weighed daily post infection. Animals were monitored three times a day for a period of 10 days post-infection. Disease progression was assessed by assigning clinical scores without knowledge of the individual genotypes: 0: Healthy; 1: mild clinical signs; 2: Up to 2 moderate clinical signs; 3: up to 3 moderate signs. Animals were culled when they showed >25% bodyweight loss or reach score 3. The most frequently observed clinical signs were piloerection, hunched posture, tremor, and labored breathing.

At the study end-point, animals were culled and terminal blood samples collected.

**Bone marrow transfer**. A single cell suspension of p110δ[E1020K-B] and wild-type (C57Bl/6.SJL) donor bone marrow was prepared in sterile HBSS (Sigma-Aldrich) as described below (Isolation of immune cells from mouse tissues). Cells from two sex-matched donors were pooled. RAG2[−/−] recipient mice were supplied with 4 mg/mL Neomycin (Sigma-Aldrich) in drinking water before sub-lethal irradiation (one dose of 500Rads over 63 s), and received $3 \times 10^6$ donor cells by intravenous (tail vein) injection. Neomycin treatment was maintained for 4 weeks post-transfer, and after 8 weeks reconstitution was confirmed by analyzing tail bleeds.

**Vaccination**. For vaccine studies, mice were immunized with Pneumovax II (pneumococcal polysaccharide vaccine, serotypes: 1–5, 6B, 7F, 8, 9N, 9V, 10A, 11A, 12F, 14, 15B. 17F, 18C, 19F, 19A, 20, 22F, 23F, and 33F; Sanofi Pasteur MSD). One 0.5 mL dose containing 50 µg/mL of each serotype was diluted 1:12.5 in sterile PBS to 4 µg/mL. Mice were given one 100 µL (0.4 µg) dose by intraperitoneal injection, 14 days before infection with *S. pneumoniae*. Blood samples were taken from the tail vein prior to vaccination and 14 days after vaccination into serum collection tubes (BD microtainer, SST) samples were centrifuged at 15000xg for 5 min and the serum stored at −20 °C until analysis.

**Nemiralisib and anti-IL-10 treatment**. The PI3Kδ inhibitor nemiralisib (GSK2269557) was supplied by GSK, Respiratory Refractory Inflammation DPU, UK. The nemiralisib suspension was prepared on the day of use in 0.2% Tween80, and administered intranasally to mice twice daily at 0.2 mg/kg in a total volume of 50 µL, as described above (*S. pneumoniae* infections). Unless otherwise specified, the first dose was given 24 h before infection, and treatment was maintained for the duration of the study. Anti-IL-10 (JES5-2A5) and anti-HRPN rat IgG1 was purchased from BioXcell and administered intranasally once daily as described above at 200 µg/50 µL. Treatment was started 24 h before infection and maintained for the duration of the study

**Isolation of immune cells from mouse tissues**. Mice were euthanized by $CO_2$ inhalation and cervical dislocation. Lungs were perfused with 10 mL cold PBS through the right ventricle, and collected into cold PBS. Lungs were homogenized using a GentleMACS tissue dissociator and mouse lung dissociation enzyme kit from Miltenyi according to the manufacter's instructions. The homogenate was transferred to 15 mL tubes (BD Falcon) and washed by centrifugation (500xg) in 10 mL cold PBS. The pellet was resuspended in 3 mL 37.5% isotonic Percoll (Sigma) at room temperature and centrifuged at 650xg for 20 min with low acceleration and no brake. The supernatant including tissue debris was removed, and the cell pellet was washed and resuspended in cold PBS. Where CFU counts were required, the right lung was processed as described above and the left lung was processed for CFU counts as described below. Bone marrow cells were

**Fig. 8** p110δ[E1020K] mice have an expanded population of IL-10 producing CD19[+]B220[−] B cells. **a** Naïve p110δ[E1020K] mice show a significant increase in the proportion and number of B220[−]B cells in bone marrow and spleen. While this population is detectable in wild-type mice, it is absent from p110δ[D910A] mice. Representative pseudocolor plots show mean cell proportions in naïve mice **b** *S. pneumoniae* (S.p) infection does not lead to significant expansion or recruitment of B220[−] B cells at 24 h post-infection in bone marrow, blood, spleen or lung tissue. Representative pseudocolor plots show mean cell proportions in naïve mice. **c** At 24 h post *S. pneumoniae* infection, IL-10 production is increased in p110δ[E1020K] B cells from both lung and spleen, and reduced in p110δ[D910A] B cells compared to wild-type mice. Furthermore, B220[−]B cells produce higher levels of IL-10 in response to S. pneumoniae infection compared to B220[+] B cells in wild-type and p110δ[E1020K] mice. Representative pseudocolor plots show mean cell proportions in *S. pneumoniae* infected mice (Mean cell counts are shown, data-points represent individual animals. **a** Combined data from two independent experiments, WT $n = 10$; E1020K $n = 12$; D910A $n = 7$, **b** representative data from two independent experiments, $n = 6$, **c** representative data from two independent experiments $n = 6$)

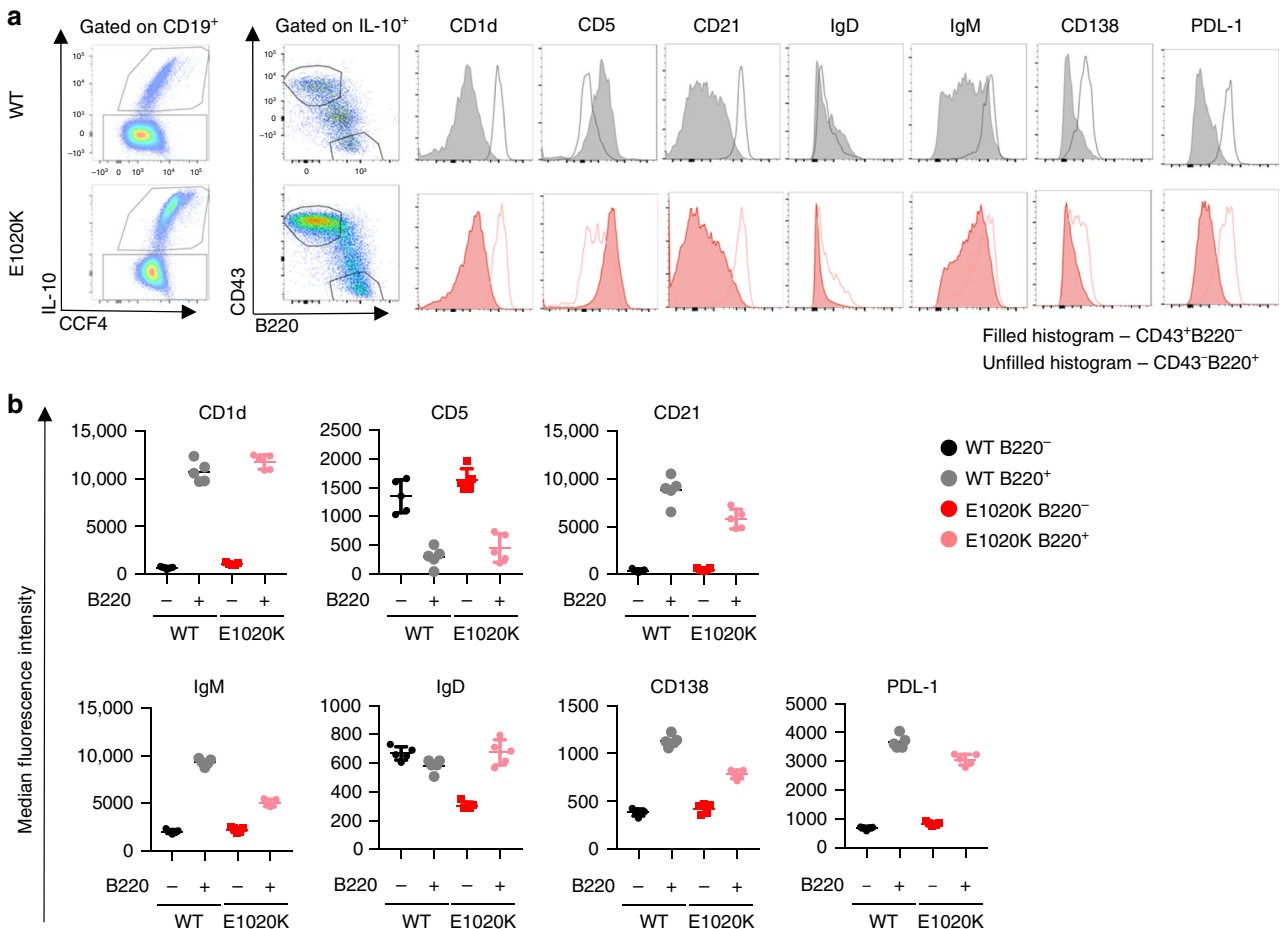

**Fig. 9** IL-10 producing B220⁻B cell subset is a novel B cell subset, expanded in PI3Kδ^E1020K mice. In order to compare surface marker expression from B220⁻ and B220⁺ IL-10 producing B cells, splenocytes from *Il10*^ITIB mice (WT and p110δ^E1020K) were stimulated with LPS/PdBu/Ionomycin/Brefeldin A for 5 h and then stained for cell surface markers as shown. **a** After gating on IL-10 producing B cells, CD43^++B220⁻ cells were compared to CD43⁻B220⁺ cells. Histograms highlighting the differential surface marker expression as measured by median fluorescence intensity (MFI) between these populations are shown. **b** Comparison of surface marker MFI show that CD19⁺ B220⁻ IL-10⁺ B cells express: CD43^++CD5^int/+ CD23⁻CD21⁻CD1d^lo/int IgM^+/-IgD^lo/- PDL1⁻CD138⁻ as opposed to conventional Bregs expressing: CD19^hi B220^hi IL-10⁺ CD43⁻ CD5^VarCD23⁻CD21^++CD1d^hi IgM^hi IgD^Var/lo PDL1⁺CD138^int. (mean MFI ± SD is shown, representative data from two independent experiments, n = 5; data-points represent individual animals)

collected by flushing cold PBS through femurs and tibias collected, filtered through 40 μm cell strainers and washed by centrifugation in 5 mL PBS. Spleens, thymus and peripheral lymph nodes were homogenized in PBS by pushing the tissue through 40 μm cell strainers (BD) using a syringe plunge. The cell suspension was then transferred to a 15 mL Tube (BD falcon) and washed once in 5 mL cold PBS. For blood, bone marrow and spleen, red blood cells (RBC) were lysed using hypotonic ammonium chloride RBC lysis buffer (Sigma). Lysing was quenched with cold PBS and the cells were collected by centrifugation. Single cell suspensions were processed for flow cytometry as described below.

***S. pneumonia*** **CFU counts.** Lungs were homogenized in 1 ml PBS using a Bullet Blender using 3 × 3 mm steel beads: speed 8, 3 min (Next Advance, USA). Spleens were homogenized as described above in 2 mL PBS. Serial dilutions (10-fold) were performed for spleen and lung homogenates, and samples were plated out on blood agar as described above. Plates were incubated for 24 h at 37 °C and *S. pneumonia* colonies were counted.

**In vitro stimulation of mouse B cells.** Splenocytes were isolated from naïve wild-type, p110δ^E1020K-GL and p110δ^D910A mice that had been crossed with the *Il10*^ITIB reporter line as described above and the total cell count was obtained using a CASY counter. The cells were resuspended in complete RPMI (RPMI plus 5% (v/v) FCS, 50 μM β-mercaptoethanol and 100 μg/ml penicillin and streptomycin) and plated at 5 × 10⁶ cells in 100 μL per well in a 96-well U bottom plate. The cells were stimulated with 10 ng/mL LPS, 50 ng/mL PdBu (Sigma, USA), 0.25 μg/mL Iono-mycin (Sigma, USA) and 1 μL/mL of Brefeldin A (eBioscience) for 5 h at 37 °C. The cells were then processed for IL-10 detection and flow cytometry as described below.

**Detection of IL-10 in *Il10*^ITIB reporter mice.** Single cell suspensions from *S. pneumoniae* infected mice or in vitro stimulated cells were prepared as described above. The cells were incubated with 3.3 μM CCF4-AM, a Fluorescence Resonance Energy Transfer substrate for β-lactamase (LiveBLAzer kit, Invitrogen), and 3.6 mM probenecid (Sigma) in complete RPMI for 90 min at 29 °C, as previously described[28], then placed on ice and collected by centrifugation. Cells were then processed for flow cytometry as described below.

**Flow cytometry.** Antibodies used for flow cytometry are listed in Table 1. Single cell suspensions were stained using an antibody master mix in PBS/0.5%BSA for 40 min at 4 °C. Cells were washed and fixed in 4% paraformaldehyde (Biolegend) for 10 min at room temperature before washing 2x in PBS/0.5%BSA. For intra-cellular detection of IL-10 in human samples, the cells were fixed and permeabi-lized with Cytofix/Cytoperm buffer (BD Biocsiences, USA) following surface staining. For FoxP3 staining, the eBioscience FoxP3/Transcription Factor staining Buffer Set was used according to the manufacturer's instructions. Non-fluorescent counting beads (AccuCount Blank Particles 5.3 μm; Spherotech, USA) were added to quantify absolute cells numbers. Samples were kept at 4 °C until analysis (BD Fortessa5). Analysis was carried out using FlowJo (Treestar) analysis software.

**In vitro stimulation of human B cells.** Freshly isolated PBMCs from patients and control individuals were stimulated with 0.5 μg/mL purified plate bound anti CD3 monoclonal antibody (clone-OKT3) (Invitrogen) and 20 ng/mL recombinant human IL2 in presence or absence of, the p110δ inhibitor, nemiralisib (10 nM) for 72 h. This stimulation led to the activation of CD40L (CD154) on CD4 T cells. The interaction between CD40L on CD4 T cells with CD40 on B cells resulted in the production of IL-10 in regulatory B cells. For a final stimulation 50 ng/mL PdBu (sigma, USA), 0.25 μg/mL ionomycin (Sigma, USA) and 1 μL/mL of Brefeldin A

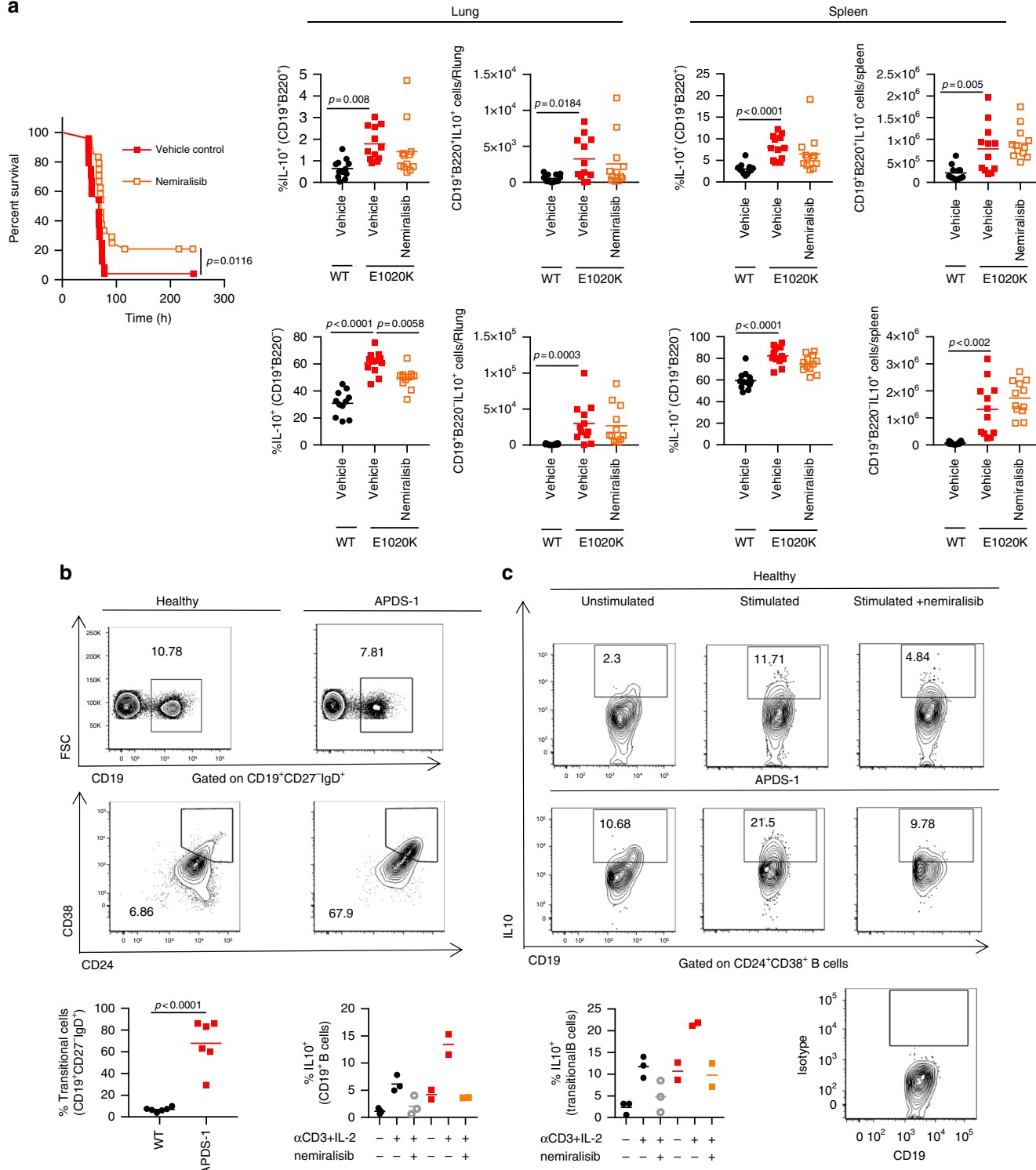

**Fig. 10** Nemiralisib treatment reduces the proportion of IL-10 producing B cells in mice and humans. **a** Prophylactic treatment with nemiralisib improved survival in p110δ$^{E1020K-GL}$ mice and was associated with a significant reduction in the proportion of IL-10 producing B220$^-$ B cells in the lung at 24 h post-infection. (**a** Combined data from two independent studies, survival $n = 24$; lung tissue analysis $n = 12$, mean cell number shown, data-points represent individual animals) **b** Blood from patients with APDS ($n = 6$) and healthy controls ($n = 6$) was obtained and the B cell phenotype was determined by flow cytometry. Transitional B cells are identified as CD19$^+$IgD$^+$CD27$^-$CD24$^+$CD38$^+$. Representative contour plots with outliers show the mean proportions of cells. **c** Freshly isolated PBMCs were unstimulated or stimulated with plate bound anti-CD3 and anti-IL-2 for 72 h in the presence or absence of nemiralisib ($n = 2$–3). The proportion of IL-10 producing cells among the total B cell and transitional B cell populations was determined by flow cytometry. Representative contour plots with outliers show mean proportions of IL-10 producing cells. (Mean cell proportions are shown, data-points represent individual patients; Combined data from two independent experiments)

**Table 1 Antibodies**

| Anti-mouse | Clone | Dilution | Supplier | Cat. no |
|---|---|---|---|---|
| CCR2 | 475301 | 1:100 | R&D Systems | FAB5538A-025 |
| CD11b | M1/70 | 1:1000 | Biolegend | 101251 |
| CD11c | N418 | 1:500 | Biolegend | 117348 |
| CD138 | 281-2 | 1:100 | Biolegend | 142503 |
| CD19 | 1D3 | 1:300 | BD Biosciences | 564296 |
| CD1d | 1B1 | 1:100 | eBioscience (ThermoFisher) | 46-0011-82 |
| CD21 | 7G8 | 1:200 | BD Biosciences | 747763 |
| CD23 | B3B4 | 1:100 | Biolegend | 101614 |
| CD25 | PC61 | 1:300 | Biolegend | 102051 |
| CD3 | 145-2C11 | 1:300 | Biolegend | 100351 |
| CD4 | GK1.5 | 1:300 | Biolegend | 100449 |
| CD43 | S7 | 1:100 | BD Biosciences | 560663 |
| CD44 | IM7 | 1:500 | Biolegend | 103032 |
| CD45 | 30-F11 | 1:300 | Biolegend | 103139 |
| CD45R (B220) | RA3-6B2 | 1:300 | BD Biosciences | 563793 |
| CD5 | 53-7-3 | 1:300 | Biolegend | 100627 |
| CD62L | MEL-14 | 1:500 | Biolegend | 104408 |
| CD8a | 53-6.7 | 1:300 | Biolegend | 100759 |
| FoxP3 | FJK-16s | 1:200 | eBioscence (ThermoFisher) | 17-5773-80 |
| IgD | 11-26c.2a | 1:400 | BD Biosciences | 564274 |
| IgM | II/41 | 1:400 | eBioscence (ThermoFisher) | 61-5790-82 |
| IL-7R | A7R34 | 1:200 | Biolegend | 135014 |
| KLRG1 | 2F1 | 1:300 | Biolegend | 138416 |
| Ly6C | HK1.4 | 1:500 | Biolegend | 128024 |
| Ly6G | 1A8 | 1:500 | Biolegend | 127633 |
| Ly6G/Ly6C (Gr1) | RB6-8C5 | 1:500 | Biolegend | 108441 |
| NK1.1 | PK136 | 1:300 | Biolegend | 108728 |
| PDL1 | 10F.9G2 | 1:400 | Biolegend | 124319 |
| Siglec-F | E50-2440 | 1:500 | BD Biosciences | 552126 |
| TCRβ | H57-597 | 1:300 | Biolegend | 109224 |
| γδTCR | UC7-13D5 | 1:100 | Biolegend | 107504 |
| *Anti-human* | | | | |
| CD19 | SJ25C1 | 1:100 | BD Biosciences | 563326 |
| CD24 | ML5 | 1:100 | BD Biosciences | 562789 |
| CD38 | HIT2 | 1:50 | BD Biosciences | 555459 |
| IL-10 | JES3-9D7 | 1:12.5 | Miltenyi Biotec | 130-096-043 |

(eBioscience) were also added for the last 4 h. Cells were washed, surface stained, fixed and permeabilized with Cytofix/Cytoperm buffer (BD Biocsiences, USA) for the intracellular detection of IL-10 as described above.

**Natural antibody ELISA.** Anti-phosphorylcholine IgM and IgG levels were assessed in serum from naïve mice. Blood samples were collected by cardiac puncture and serum collected as described above. NUNC Maxisorp ELISA plates were coated overnight at 4 °C with 20 μg/mL phosphorylcholine-BSA (BioSearch) in sodium bicarbonate coating buffer (pH9) (Biolegend) (100 μL/well). Plates were washed 4× with PBS/0.2% Tween20 followed by blocking for 1 h at room temperature in PBS/1%BSA. Serum samples were diluted 1:10 in PBS/1% BSA and pooled WT serum was diluted from 1:5 to 1:400 to confirm the linear range of the assay. After removing the blocking solution, samples were added at 100 μL/well, and the plates were incubated overnight at 4 °C. The plates were then washed 4x in PBS/0.2%Tween20. Polyclonal HRP conjugated goat-anti-mouse IgM (abcam ab97023) or IgG (abcam ab97230) was diluted 1:5000 in PBS/1% BSA, added at 100 μL per well and the plates were incubated for 2 h at room temperature. The plates were washed 6× with PBS/0.2%Tween20, TMB substrate (Biolegend) was added at 100 μL per well and incubated for 5–10 min at room temperature. The reaction was stopped by adding 50 μL 2 N $H_2SO_4$ solution. Absorbance was read at 450 nm (Fluostar omega, BMG Labtech).

**Tissue resident cell analysis (lung).** Mice received an intravenous (tail vein) injection of 3 μg anti-mouse CD45 conjugated to biotin (Biolegend, clone 30-F11)

in 100 μL PBS. Four minutes after injection animals were killed and the lungs were collected into cold PBS without prior perfusion. Lungs were then finely minced using a scalpel blade and the pushed through a 100 μm cell strainer (BD) the homogenate was collected in a 15 mL tube (BD falcon) and washed in 10 mL cold PBS. The pellet was resuspended in 3 mL 37.5% isotonic Percoll (Sigma) at room temperature and processed as described above. The cells were stained for flow cytometry as described above, including streptavidin APC to identify anti-CD45-Biotin labeled circulating cells.

**Cytokine analysis.** Cytokine levels were measured in the supernatant from lung homogenates prepared for CFU analysis. The homogenates were centrifuged for 1 min at 10,000×g to remove tissue debris. Samples collected from wild-type mice treated with nemiralisib were analyzed using Meso Scale Discovery mouse Th1/Th2 9-plex ultrasensitive kits, according to manufacturer's instructions. Samples collected from genetically modified mice were analyzed using the flow cytometry based Legendplex mouse inflammation 13-plex panel (Biolegend).

**Gene arrays.** Wild-type mice were treated with nemiralisib and infected with *S. pneumoniae* as described above. At 24 h post-infection, whole lungs were collected and snap frozen in liquid nitrogen. Samples were randomized and analyzed (Affymetrix Genechip Mouse genome 430 2.0 Array) by Expression Analysis, Quintiles Global Laboratories. Data analysis was performed by Computational Biology and Statistics, Target Sciences, GlaxoSmithKline, Stevenage. Data was normalized using the robust multiarray average method[46] and quality checked in R/Bioconductor[47] using the *affy* package[48]. A linear model was fitted to the RMA normalized data for each probset and differential expression analysis was conducted in the ArrayStudio software (Omicsoft Corporation). *p*-values were false discovery rate corrected by the method of Benjamini and Hochberg[49]. Probes with an absolute fold change >1.5 and an adjusted *p*-value < 0.05 were called significant.

**Luminex antibody analysis.** Measurement of IgG recognizing pneumococcal polysaccharide of 13 serotypes was performed as described previously[50]. The assay was modified in order to measure murine IgG as follows: 50 μL of 1:10 diluted mouse sera was added to each well containing the pneumococcal polysaccharides coupled to microspheres and the secondary antibody used was 50 μL of 10 μg/mL goat F(ab')₂ anti-mouse IgG (H + L)- R-phycoerythrin (Leinco Technologies, Inc). R-PE fluorescence levels were recorded for each serotype and median fluorescence intensities were compared for the different genotypes.

**PIP₃ quantification.** CD4 and CD8 cells were isolated from mouse splenocytes using immunomagnetic negative selection by biotinylated cocktail of antibodies and Streptavidin dynabeads (Invitrogen). T cells were stimulated with anti-CD3 (1 mg/mL), (145-2c11, Biolegend) anti-CD28 (2 μg/mL), (35.51, Biolegend) anti-bodies followed by crosslinking with anti-Armenian hamster IgG (10 μg/mL, Jackson ImmunoResearch labs). Cells were stimulated for 1 min at 37 °C. B cells were similarly isolated from mouse splenocytes and stimulated for 1 min at 37 °C with anti-IgM 2 μg/mL (AffinPure F(ab')₂ Fragment goat anti-mouse IgM from Jackson Immuno-Research labs).

After stimulation of the cells, we terminated the reactions by addition of 750 μL kill solution (CHCl₃:MeOH:1 M HCL (10:20:1) and immediately froze the samples on dry ice. PIP₃ levels were quantified by mass spectrometry as previously described[51].

**Western blotting.** Purified T and B cells were isolated from murine spleens and stimulated as described above (PIP₃ measurement). For performing western blots the stimulation period was 5 min. Isolated T cells were stimulated with anti-CD3 1 μg/mL and anti-CD28 (2 μg/mL) with or without 10 Nm nemiralisb; or B cells with anti-IgM 4 μg/mL (AffinPure F(ab')₂ Fragment goat anti-mouse IgM from Jackson ImmunosResearch labs).

Stimulated T and B cells were lysed with ice-cold lysis buffer (50 mM HEPES, 150 mM NaCl, 10 mM NaF, 10 mM Indoacetamide, 1% IGEPAL and proteinase inhibitors (Complete Ultra tablets, Roche)) for 15–20 min. Lysates were centrifuged at 15,000×g for 10 min at 4 °C and supernatants were mixed with NuPage LDS sample buffer (life technologies). Samples were heated at 70 °C and resolved on 4–12% NuPage bis-tris gel (Invitrogen), transferred to PVDF membranes and probed with the following antibodies: pAKT (T308, Cell Signaling, 1 in 1000 dilution); total AKT1 (2H10, Cell signaling, 1 in 2000 dilution); p110δ (Sc7176, Santa Cruz Biotechnology, 1 in 200 dilution); pS6 (S235/236, Cell Signaling, 1 in 500 dilution); pFoxo1/3a (T24/T32, Cell Signaling, 1 in 1000 dilution); pErk (p44/42, Cell Signaling, 1 in 200 dilution); βActin (Sc47778, Santa Cruz Biotechnology, 1 in 2000 dilution).

## Statistics and general methods

**Sample selection.** For animal studies where disease phenotype is the outcome we determined that a minimum of 10–12 mice per group is required to detect a 15% increase in survival with 80% power with a two-sided significance of 0.05. This result is based

on preliminary experiments showing a median survival time of 50 h. In experiments where analysis of different cell subsets or gene expression analysis is the main end point, groups of 6–8 mice is sufficient. Regardless of the cohort size, experiments were repeated at least once in order to determine inter-experimental variability as well as inter individual variability. We have arrived at these numbers in consultation with statisticians at the Babraham Institute and at GSK.

Samples were excluded from analysis only when there was a known technical problem affecting the analysis. For in vivo studies, genetically modified mice and wild-type mice were age and sex matched. PI3Kδ inhibitor and vaccine treatment groups were also age and sex matched to vehicle control groups. Wild-type littermates were from p110δ$^{E1020K}$ colonies and were co-housed with p110δ$^{E1020K}$ mice. For other strains (e.g., p110δ$^{D910A}$), p110δ$^{E1020K}$ littermates or C57bl/6 mice from the Babraham Institute breeding colony were used as controls. Gene array samples were randomized by GSK statisticians prior to shipping for mRNA extraction by Expression Analysis, Quintiles.

**Blinding**. In vivo studies were blinded as follows: animals were genotyped and allocated to age and sex matched treatment groups. Pneumovax immunization was performed by technicians not involved in study design, analysis or interpretation. The genotype and vaccine treatment information was hidden on the experimental study sheets and mouse cages. Mice were identified by implanted microchips, so the identity was not apparent to operator. Intranasal infections and monitoring were done without knowledge of genotypes. Genotype and vaccine treatment group information was only revealed at the study endpoint. Monitoring in PI3Kδ inhibitor treatment studies was not fully blinded due to practical constraints: the researcher performing the intranasal treatment also prepared the compound formulation. However, genotypes were blinded in these studies as described above, and intranasal infections were performed without knowledge of genotype or PI3Kδ inhibitor treatment group. For all studies, monitoring was shared between 1−2 researchers and 3 technicians not responsible for the design, analysis or interpretation of the study. In vitro studies were not blinded.

**Data analysis**. Data analysis was performed in Graphpad Prism. We performed a D'Agostino & Pearson normality test for sample sizes of $n > 8$, and a Shapiro–Wilk normality test for sample sizes of $n < 8$. For data sets following a Gaussian distribution, we used student's $t$-test with Welch's correction to account for samples with unequal variance where two groups were compared and one-way ANOVA with Tukey's multiple comparisons test where three or more groups were compared. For data sets that did not follow a Gaussian distribution we performed the following non-parametric tests: where two groups were compared we used the Mann–Whitney test. Where three or more groups were compared, we used the Kruskal–Wallis test with Dunn's multiple comparison test. All tests were two-sided without matching or pairing of values to be compared. Survival data was analyzed using the Gehan-Breslow-Wilcoxon test. $p$-values are indicated in individual figures, and in all cases we considered results statistically significant when $p \leq 0.05$.

**Data availability**. Gene array data have been deposited in the National Center for Biotechnology Information Gene Expression Omnibus (GEO) and are accessible through GEO series accession number GSE109941. All data is available from the authors on request.

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

## Acknowledgements
We thank Rahul Roychoudhuri, Bart Vanhaesebroeck and Martin Turner for their invaluable advice on the draft manuscript. We are grateful to Jeremy Brown for *S. pneumoniae* TIGR4 and helpful discussions. We thank Hicham Bouabe and Jürgen Heesemann (LMU Munich) for providing the *Il10*^ITIB mice and for advice. We thank Ramkumar Venigalla for advice on B cell phenotyping and Alice Denton for advice on the detection of tissue resident cells in the lung. Rainer Döffinger provided advice on the use of Luminex for the quantification of antibodies against pneumococcal serotypes; Keith Burling helped quantify mouse serum immunoglobulins. Anne Sigonds-Pichon provided expert advice with regards to sample size calculation and statistical analysis. We gratefully acknowledge the support from the Babraham Institute Biological Services Unit, Biological Chemistry, Mass Spectroscopy and Flow Cytometry Facilities. Funding for the project was from the Medical Research Council MR/M012328/2 (A.S., A.M.C., S.N., K. O.), Wellcome Trust 103413/Z/13/Z and 206618/Z/17/Z (A.C.), 095691/Z/11/Z (K.O.), 095198/Z/10/Z (S.N.) Biotechnology and Biological Sciences Research Council BBS/E/B/ 000 -C0407, -C0409, -C0427 and -C0428 (K.O.). E.B.H. is currently employed by Cambridge University Hospitals NHS Foundation Trust but is seconded to spend 50% of his time on GSK clinical trial research. He receives no other benefits or compensation from GSK. E.B.H. was funded by the Wellcome Trust Translational Medicine and Therapeutics (TMAT) PhD program. S.N. is also supported by the National Institute for Health Research (NIHR) Cambridge Biomedical Research Center. M.R.C. is supported by the National Institute of Health Research (NIHR) Cambridge Biomedical Research Center and the NIHR Blood and Transplant Research Unit and by a Medical Research Council New Investigator Research Grant (MR/N024907/1) and an Arthritis Research UK Cure Challenge Research Grant (21777).

## Author contributions
Performed experiments: A.S., A.C., K.C., R.A., V.C., J.C., E.B.H. Analyzed data: A.S., A. C., K.C., R.A., V.C., J.C., S.S., G.B., K.O. Bioinformatics: G.B. Provided essential materials: A.G.R., N.H., E.M.H. Designed and supervised study: S.S., M.R.C, S.N., N.H., E.M. H., A.M.C. Wrote paper, A.S., A.C., A.M.C., and K.O., with input from E.B.H., M.R.C., S. N., and E.M.H. All authors commented on and approved of the draft manuscript before submission.
