## [Peer Review File · Nature Communications]

Reviewers' comments:

Reviewer #1 (Remarks to the Author):

A well done and interesting paper suggesting there is a novel type of B cell population with Breg characteristics in the lung that the PI3K δ hyper-activation mutation increases in number and this causes susceptibility to pneumococcal infection. Overall the data are convincing and experiments well done, and succinctly described in an easy to follow way. I do have some points / issues to raise:

Major points:

1. I really think the data need an infection experiment in the knockin mice v wild-type experiment with IL10 depletion via antibody; this is necessary to directly link the new type of Breg population to the increased susceptibility to pneumococcus. An alternative would be to do this in B cell depleted mice using antiCD20 antibody or similar if this population expresses this.

2. most of the difference in survival occurs after 24 hours (I think, difficult to tell as the x axis goes out to 700 hours) and there is no clear mechanism in the paper as to why survival is different; except the interesting data for the Ighmtm1 mice. So I do think we need bacterial CFU and histology for (a) later time point(s) when the survival curve is clearly different, plus cytokine levels just to get a feel for what is happening. The histology is especially important given that the mechanism should be some sort of immuno-dampening effect. Although it will probably take more experiments than this to know for sure what the mechanism of improved survival is, I think that is beyond the scope of this paper; just some data to give a feel for what is happening would be important otherwise it is all a bit too mysterious

Minor points:

3. a lot of supplementary fig 7 I think needs to be in the main paper: bacterial CFU at 24 hours, total IL10, cell specific IL10 especially the B cell data), effects of nemiralisib on IL10 levels and the volcano plot. It looks like the total IL10 levels are not much affected which weakens their case a bit (but timing / localisation etc may make it hard to find a clear difference in IL10 in lung homogenates)

4. the x axis for the survival curves should be shortened / made into a broken x axis as the tail of parallel survival does not help and we need the detail for the earlier timepoints.

5. Fig. 1 S pneumoniae strain type and inoculum size need stating as it does in other infection experiments

6. line 145 re. p110 δ D910A mice not having increased susceptibility, in these experiments the mice immune are naive to pneumococcus so we would not expect antibody to be protective, apart from natural IgM

7. the protective effect of T cell expression of p110 δ E1020K is very interesting any explanation for this effect? Also this protective effect could have cancelled out the susceptibility of B cell p110 δ E1020K mice; yet in fact the all cell p110 δ E1020K knock out has similar phenotype to the B cell KOs alone. Any suggestions as to why the B cell defect is dominant for the phenotype of the total KO mouse? Some thoughts in the discussion line 306 would be useful

8. the exclusion of natural IgM as a mediator for the increased susceptibility would be strengthened if the authors measured IgM recognition of S. pneumoniae in sera taken from uninfected mice using flow cytometry

9. the Ighmtm1 phenotype is very interesting; again further development of this observations is probably beyond the scope of this paper

Reviewer #2 (Remarks to the Author):

The severity of pneumococcal infections depends largely on host's susceptibility. Here, the authors indicated in a comprehensive study the a subset of B cells (CD19+B220-) induced by hyperreactivity of PI3Kδ. Upon this induction the host - here mouse in experimental designs and in a translational study patients suffering on APDS – are more susceptible against pneumococcal infection. Importantly, the translational and therapeutic aspect could be considered here, as a PI3K inhibitor –nemiralisib – was used and the results indicated that the inhibitor protects against fulminant pneumococcal infections. Noteworthy, the population of CD19+B220- cells secretes IL-10 in an expanded manner, which is consequently reduced by nemiralisib. However, the increased susceptibility due to the expansion of the B cell population occurs only in the first stages of the infection.

In conclusion, this is an excellent study, perfectly designed and carried out and of broad interest as well.

specific comments:

- the experiments are described in an excellent way and all information needed are provided. Here, it is also important to mention that the authors have not only provided the names of antibodies used but also all the other information needed to repeat the experiments.
- the study defined the cell populations and IL-10 producing cells. Because not all readers are immunologists, the author can add some more information about the gating strategy engaged by their FACS analysis and in addition, explain the values (mostly as %) given as insets in the histograms or dot plots
- Figure 2C and D: the Western blots are shown below the panel for stimulated and treated cells. However, one can only speculate that the blots are shown for stimulated cells and not for nemiralisib treated cells. Please clarify.
- The description of figure 3B is, at least for the reviewer, confusing. Please rephrase and improve the description of the data shown in Fig. 3B. Does n=5 mean 5 mice?
- the labeling of some figures (e.g. as in Fig. 2, 7, 8 and supplementary figures) have to be explained in a better way and without repetitions. The meaning of the – and + is not always easy to follow and the symbols can be mentioned simply in the legend.
- the WHO reference (ref 1) has to be updated

The introduction is clear and focused. However, part of the introduction summarizes again the results of the study (starting in line 54). This is nice, but unnecessary. This part can simply be shortened and if desired, in parts shifted to the discussion.

Reviewer #3 (Remarks to the Author):

Overall this is a very interesting manuscript. Knockin mice with the p110 delta E1020K mutation equivalent to that seen in patients with CVID present with an expanded B220-CD19+ population which expresses IL-10 and also infiltrates the lungs. These cells are assumed to be transitional B cell-like and B-reg like cells and the authors implicate them in the susceptibility to more severe S. pneumoniae infection seen in these mice. There is considerable enthusiasm for these extensive studies but there are some relatively easily filled lacunae that should be addressed. The evidence that the B220-CD19+ population is pathogenic is indirect but the phenotype of these cells could be clarified in a couple of ways

a) Do the CD19+B220- B cells express the AA4.1/CD93 marker at levels comparable to wild type murine transitional B cells?

b) To partially rule out the possibility that these cells are generated primarily in a hyperactive T-cell dependent fashion, the frequency of these B220-CD19+ B cells should have been examined in the Mb-1 Cre version of the E1020K p100delta knockin mouse that the authors describe. It is formally possible that these B cells are not transitional but more plasmablast-like or activated B cells.

c) The discussion should make it clear that it has not been firmly established that the B200-CD19+ cells are pathogenic, though they well might be

Shiv Pillai

Reviewers' comments:

Reviewer #1 (Remarks to the Author):

A well done and interesting paper suggesting there is a novel type of B cell population with Breg characteristics in the lung that the PI3K δ hyper-activation mutation increases in number and this causes susceptibility to pneumococcal infection. Overall the data are convincing and experiments well done, and succinctly described in an easy to follow way. I do have some points / issues to raise:

Major points:

1. I really think the data need an infection experiment in the knockin mice v wild-type experiment with IL10 depletion via antibody; this is necessary to directly link the new type of Breg population to the increased susceptibility to pneumococcus. An alternative would be to do this in B cell depleted mice using antiCD20 antibody or similar if this population expresses this.

Since IL-10 is a key regulatory cytokine and previous work showed that a complete lack of IL-10 leads to increased mortality following *S. pneumoniae* infection, we studied the effect of partial/local IL-10 depletion via the following two strategies.

1. Homozygous *Il10*^{IT1B} mice are IL-10 hypomorphic, therefore, we used heterozygous reporter mice for IL-10 detection. However, we exploited this property to achieve reproducible systemic downregulation of IL-10 levels. Infection of *Il10*^{IT1B} homozygous p110 δ ^{E1020K} and WT mice as well as *Il10*^{IT1B} WT p110 δ ^{E1020K} and WT mice revealed reduced serum IL-10 levels in *Il10*^{IT1B} homozygous mice at death (supplementary fig 7A). However, reduced systemic IL-10 levels did not alter the survival outcome of p110 δ ^{E1020K} or WT mice compared to *Il10*^{IT1B} WT mice (supplementary fig 7B).
2. We considered that while systemic IL-10 may be protective, high levels of local IL-10 in the lung environment during early stages of infection could be detrimental. Therefore, we treated WT mice with daily intranasal anti-IL-10 JES5-2A5 (200 μ g). These studies showed a non-significant trend towards improved survival (supplementary fig 7C). However, when we depleted IL-10 from B cell conditional p110 δ ^{E1020K-B} mice, they showed increased mortality compared to isotype control treated mice supplementary fig 7D.

These results lead us to conclude that, while IL-10 production is a key characteristic of the B220⁺ B cell population described here, it is not the main/only mechanism whereby these cells contribute to pathology during early *S. pneumoniae* infections.

See text added to line 274 (results) and line 345 (discussion)

2. most of the difference in survival occurs after 24 hours (I think, difficult to tell as the x axis goes out to 700 hours) and there is no clear mechanism in the paper as to why survival is different; except the interesting data for the Ighmtm1 mice. So I do think we need bacterial CFU and histology for (a) later time point(s) when the survival curve is clearly different, plus cytokine levels just to get a feel for what is happening. The histology is especially important given that the mechanism should be some sort of immuno-dampening effect. Although it will probably take more experiments than this to know for sure what the mechanism of improved survival is, I think that is beyond the scope of this paper; just some data to give a feel for what is happening would be important otherwise it is all a bit too mysterious.

We chose the 24h time-point for CFU/cytokine analysis, because it is just before the mice start showing clinical signs. We believe the increased susceptibility of p110 δ ^{E1020K} mice is due to a mechanism at work during the first hours after infection. For instance, the inhaled p110 δ inhibitor nemiralisib had no effect when administered as soon as 8 hours after infection, but did improve disease outcome when administered from 24h prior to infection. Some animals, especially from the p110 δ ^{E1020K} groups reach the humane endpoint from 48h onwards. This complicates the interpretation of results at later timepoints, because study groups contain individuals with a wide variation in health status which could affect experimental readouts independently of genotype. In addition, analysis at later timepoints will invariably select for individuals surviving to that timepoint. We have analysed lungs from mice by immunohistochemistry, but have not observed changes that shed further light on the mechanism of action. So far, using histology or flow cytometry, we have not observed significant differences in inflammatory infiltrates into the lung. Hence, we prefer to focus on the data that we have included in this manuscript which clearly highlights phenotypic and functional changes as consequences of altered PI3K δ activity.

Minor points:

3. a lot of supplementary fig 7 I think needs to be in the main paper: bacterial CFU at 24 hours, total IL10, cell specific IL10 especially the B cell data), effects of nemiralisib on IL10 levels and the volcano plot. It looks like the total IL10 levels are not much affected which weakens their case a bit (but timing / localisation etc may make it hard to find a clear difference in IL10 in lung homogenates)

We have moved Supplementary figure 7 into the main text to become Figure 7. Given that the Nat Comm guidelines allow for up to 10 display items, we also moved supplementary figure 2 (B cell phenotype in p110 δ ^{E1020K-GL} mice) to the main paper (now Figure 3).

4. the x axis for the survival curves should be shortened / made into a broken x axis as the tail of parallel survival does not help and we need the detail for the earlier time points.

We have looked into this but do not find that broken axis show the results more clearly. The majority of animals are culled between 48-72h and we can show a snapshot of survival proportions at this time (See figure 1 below). However, we believe that the survival curve is more informative if it shows disease progression throughout the duration of the study (10 days). The median survival values also show the kinetics of disease progression: Wild-type 71.1h; p110 δ ^{E1020K} 58h, p110 δ ^{D910A} 113.47h, indicating that p110 δ ^{E1020K} mice have accelerated disease development.

Additional Figure 1: Snapshot of survival proportions between 48-72h post infection

Bar graph showing the proportion of surviving mice at 48h-72h post infection with *S. pneumoniae*, indicating that survival rates of p110 δ ^{E1020K} mice are reduced at early timepoints.

5. *Fig. 1 S pneumoniae strain type and inoculum size need stating as it does in other infection experiments*

We used the strain TIGR4, serotype 4. We added this information to the figure legend of Fig 1 and line 77 of the text. This is also indicated on page 5 line 144 and materials and methods (page 12, lines 412, 425).

6. *line 145 re. p110δD910A mice not having increased susceptibility, in these experiments the mice immune are naive to pneumococcus so we would not expect antibody to be protective, apart from natural IgM*

The key point of interest here is that D910A mice are devoid of so-called natural IgM antibodies as illustrated in Fig 4B and 4C. We changed the order of the graphs in the figure and moved the paragraph describing natural antibody levels to line 147 to clarify the link between disease susceptibility and natural antibody levels.

7. *the protective effect of T cell expression of p110δE1020K is very interesting any explanation for this effect? Also this protective effect could have cancelled out the susceptibility of B cell p110δE1020K mice; yet in fact the all cell p110δE1020K knock out has similar henotype to the B cell KOs alone. Any suggestions as to why the B cell defect is dominant for the phenotype of the total KO mouse? Some thoughts in the discussion line 306 would be useful.*

This is part of an ongoing study. We raise a number of possible explanations in the discussion we prefer not to speculate further until we have additional data.

8. *the exclusion of natural IgM as a mediator for the increased susceptibility would be strengthened if the authors measured IgM recognition of S. pneumoniae in sera taken from uninfected mice using flow cytometry*

Fig 3F (now fig 4 B and C) shows that anti-PC IgM antibodies are equivalent in WT and p110δ^{E1020K} mice, but absent in p110δ^{D910A} mice. Anti-PC antibodies are generally thought to provide protection against encapsulated bacteria, including *S. pneumoniae* and are present in naïve animals.

9. *the Ighmtm1 phenotype is very interesting; again further development of this observations is probably beyond the scope of this paper*

We were very surprised to find that Ighmtm1 (μMT) mice are resistant to *S. pneumoniae* mediated pathology. This is independent confirmation that B cells cause increased susceptibility to *S. pneumoniae* and that antibodies are not required for protection at early stages.

Reviewer #2 (Remarks to the Author):

The severity of pneumococcal infections depends largely on host's susceptibility. Here, the authors indicated in a comprehensive study the a subset of B cells (CD19+B220-) induced by hyperreactivity of PI3Kδ. Upon this induction the host - here mouse in experimental designs and in a translational study patients suffering on APDS – are more susceptible against pneumococcal infection. Importantly, the translational and therapeutic aspect could be considered here, as a PI3K inhibitor –nemralisib – was used and the results indicated that the inhibitor protects against fulminant pneumococcal infections.

Noteworthy, the population of CD19+B220⁻ cells secretes IL-10 in an expanded manner, which is consequently reduced by nemiralisib. However, the increased susceptibility due to the expansion of the B cell population occurs only in the first stages of the infection. In conclusion, this is an excellent study, perfectly designed and carried out and of broad interest as well.

We are very grateful for these supportive comments.

specific comments:

- *the experiments are described in an excellent way and all information needed are provided. Here, it is also important to mention that the authors have not only provided the names of antibodies used but also all the other information needed to repeat the experiments.*

Thank you for this comment. We very much hope the detailed methods will be useful to readers.

- *the study defined the cell populations and IL-10 producing cells. Because not all readers are immunologists, the author can add some more information about the gating strategy engaged by their FACS analysis and in addition, explain the values (mostly as %) given as insets in the histograms or dot plots.*

We have changed the figures to make the gating strategy more obvious. We added supplementary Figure 8 showing the full gating strategy for identifying B220⁻ B cells. We added information to clarify which populations were gated on in the figures and in the figure legends. We clarified the meaning of numbers given in pseudocolour plots in the figure legends.

- *Figure 2C and D: the Western blots are shown below the panel for stimulated and treated cells. However, one can only speculate that the blots are shown for stimulated cells and not for nemiralisib treated cells. Please clarify.*

The blots show phospho Akt in unstimulated cells, stimulated cells and stimulated cells that have been nemiralisib-treated. We agree that Fig 2A and 2B could be clearer and have changed these, but we believe that Fig 2C and 2D are clearly marked.

- *The description of figure 3B is, at least for the reviewer, confusing. Please rephrase and improve the description of the data shown in Fig. 3B. Does n=5 mean 5 mice?*

Fig 3B is a survival curve (n=22). Does the reviewer mean Fig 2B? In that figure and elsewhere in the manuscript, each data-point represents one biological replicate (e.g. one mouse). We updated the figure legends to reflect this.

- *The labeling of some figures (e.g. as in Fig. 2, 7, 8 and supplementary figures) have to be explained in a better way and without repetitions. The meaning of the – and + is not always easy to follow and the symbols can be mentioned simply in the legend.*

We have corrected this where it was unclear.

- the WHO reference (ref 1) has to be updated

We added the following reference instead of the reference to the WHO web site:

O'Brien, K. L. et al. Burden of disease caused by *Streptococcus pneumoniae* in children younger than 5 years: global estimates. *The Lancet* 374, 893-902, doi:https://doi.org/10.1016/S0140-6736(09)61204-6 (2009).

The introduction is clear and focused. However, part of the introduction summarizes again the results of the study (starting in line 54). This is nice, but unnecessary. This part can simply be shortened and if desired, in parts shifted to the discussion.

We find it is often useful to summarise the key findings at the end of the introductory paragraph, but will take the editor's advice on this.

Reviewer #3 (Remarks to the Author):

*Overall this is a very interesting manuscript. Knockin mice with the p110 delta E1020K mutation equivalent to that seen in patients with CVID present with an expanded B220-CD19+ population which expresses IL-10 and also infiltrates the lungs. These cells are assumed to be transitional B cell-like and B-reg like cells and the authors implicate them in the susceptibility to more severe *S. pneumoniae* infection seen in these mice. There is considerable enthusiasm for these extensive studies but there are some relatively easily filled lacunae that should be addressed. The evidence that the B220-CD19+ population is pathogenic is indirect but the phenotype of these cells could be clarified in a couple of ways*

a) Do the CD19+B220- B cells express the AA4.1/CD93 marker at levels comparable to wild type murine transitional B cells?

No, The CD19⁺B220⁻ B cells in both WT and p110δ^{E1020K} mice express intermediate levels of CD93, while transitional B cells are defined as CD93⁺. See additional **Figure 2** below

Additional Figure 2: Intermediate expression of CD93 on B220⁻ B cells

Representative Dot plots and histograms showing intermediate CD93 expression on CD19⁺B220⁻ B cells in the spleen of naïve WT and p110δ^{E1020K} germline KI mice. Histogram Y-axis is normalised to mode. Results representative of 5 individual animals.

b) To partially rule out the possibility that these cells are generated primarily in a hyperactive T- cell dependent fashion, the frequency of these B220-CD19+ B cells should have been examined in the Mb-1 Cre version of the E1020K p10delta knockin mouse that the authors describe. It is formally

possible that these B cells are not transitional but more plasmablast-like or activated B cells.

We added a panel to Figure 8A showing increased B220⁻ B cell numbers and proportions in the spleen of mb1^{cre} p110δ^{E1020K-B} mice and added the following text to line 214: “Analysis of B cell conditional p110δ^{E1020K-B} mice show a similar expansion of B220⁻ B cells in the spleen (Fig 8A) and bone marrow (data not shown), indicating that the development of this population in response to PI3Kδ hyperactivation is B-cell intrinsic” Figure 9 show that the B220⁻CD19⁺ B cells do not express CD138, indicating that they are not likely to be plasma blasts.

c) The discussion should make it clear that it has not been firmly established that the B200-CD19+ cells are pathogenic, though they well might be

The discussion clearly states that this is a correlation:

“We describe a subset of B cells which lack the common B cell marker B220, and whose abundance is *correlated* with the susceptibility to *S. pneumoniae* infection.”

However, reviewer 3 is correct that we have not proven a cause and effect. We have therefore altered the abstract to emphasise that the expansion of this novel B cell population *correlates* with increased susceptibility to *S. pneumoniae*. Evidence in support of this population being pathogenic is

- 1) Only p110δ^{E1020K-B} mice recapitulated the p110δ^{E1020K-gl} phenotype
- 2) CD19⁺B220⁻ B cells are increased massively in p110δ^{E1010K} mice and absent in p110δ^{D910A} mice.
- 3) CD19⁺B220⁻ were found in the lung where we suspect cells that confer susceptibility would reside.

So while we acknowledge that we are describing a correlation rather than a proven cause and effect, the corroborative evidence is strong.

We hope this changes and clarifications address any concerns by the reviewers and the editors.

REVIEWERS' COMMENTS:

Reviewer #1 (Remarks to the Author):

the answers to my comments are fine - this paper represents a lot of work and shows very interesting findings

Reviewer #2 (Remarks to the Author):

The authors have adequately responded to the reviewers' comments and additional experiments were performed to support their findings. I have no further comments on this highly interesting study.

Reviewer #3 (Remarks to the Author):

I am satisfied with the responses